



# Can terminal settling velocity and drag of natural particles in water ever be predicted accurately?

Onno J.I. Kramer[1/2/3/4], Peter J. de Moel[5], Shravan K.R. Raaghav[2], Eric T. Baars[3], Wim H. van Vugt[4], Wim-Paul Breugem[2], Johan T. Padding[2] & Jan Peter van der Hoek[1/3]

[1]Delft University of Technology, Faculty of Civil Engineering and Geosciences, Department of Water Management, PO Box 5048, 2600 GA, Delft, The Netherlands, (E-mail: o.j.i.kramer@tudelft.nl), Tel: +31 6-42147123
[2]Delft University of Technology, Faculty of Mechanical, Maritime and Materials Engineering, Department of Process and Energy, Leeghwaterstraat 39, 2628 CB, Delft, The Netherlands
[3]Waternet, PO Box 94370, 1090 GJ, Amsterdam, The Netherlands
[4]HU University of Applied Sciences Utrecht, Institute for Life Science and Chemistry, PO Box 12011, 3501 AA, Utrecht, The Netherlands
[5]Omnisys, Eiberlaan 23, 3871 TG, Hoevelaken, The Netherlands

*Correspondence to*: Onno Kramer (onno.kramer@waternet.nl)

**Abstract.** Natural particles are frequently applied in drinking water treatment processes in fixed bed reactors, in fluidised bed
reactors, and in sedimentation processes to clarify water and to concentrate solids. When particles settle, it has been found that in terms of hydraulics, natural particles behave differently when compared to perfectly round spheres. To estimate the terminal settling velocity of single solid particles in a liquid system, a comprehensive collection of equations is available. For perfectly round spheres, the settling velocity can be calculated quite accurately. However, for naturally polydisperse non-spherical particles, experimentally measured settling velocities of individual particles show considerable spread from the calculated
average values.

This work aimed to analyse and explain the different causes of this spread. To this end, terminal settling experiments were conducted in a quiescent fluid with particles varying in density, size and shape. For the settling experiments, opaque and transparent spherical polydisperse and monodisperse glass beads were selected. In this study, we also examined drinking water related particles, like calcite pellets and crushed calcite seeding material grains, both applied in drinking water softening.
Polydisperse calcite pellets were sieved and separated to acquire more uniformly dispersed samples. In addition, a wide variety of grains with different densities, sizes and shapes were investigated for their terminal settling velocity and behaviour. The derived drag coefficient was compared with well-known models such as Brown–Lawler.

A sensitivity analysis showed that the spread is caused to a lesser extent by variations in fluid properties, measurement errors and wall effects. Natural variations in specific particle density, path trajectory instabilities and distinctive multi-particle settling
behaviour caused a slightly larger degree of spread. In contrast, greater spread is caused by variations in particle size, shape and orientation.

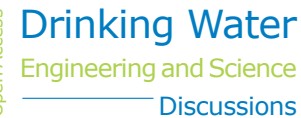

In terms of robust process designs and adequate process optimisation for fluidisation and sedimentation of natural granules, it is therefore crucial to take into consideration the influence of natural variations of the settling velocity when using predictive models for round spheres.


**Keywords**    Drinking Water; Terminal Settling Velocity; Calcium Carbonate Grains; Drag Coefficient; Natural Particles; Data Spread

# 1 Introduction

## 1.1 Deviations in the prediction of settling in water treatment processes

The settling behaviour of *natural* grains in drinking water treatment processes is of great importance (Camp, 1852); (Cheremisinoff, 2002); (Edzwald, 2011); (Howe et al., 2012); (Crittenden et al., 2012). Examples include pellet softening in fluidised bed reactors (Graveland et al., 1983), sedimentation, flotation and flocculation, filtration processes (Amburgey, 2005); (Tomkins et al., 2005), backwashing of filter media, and washing columns in which fine material and impurities are separated from seeding material (Cleasby et al., 1977); (Soyer and Akgiray, 2009). In processes such as pellet softening (Rietveld, 2005); (van Schagen, 2009), it is important always to keep the particles in fluidised state, i.e. to prevent fixed bed state (which sets the minimum superficial velocity) or flushing state (which sets the maximum superficial velocity). In contrast, in sand filter backwash processes, exceeding the maximum settling velocity, i.e. flushing of impurities and fine materials, is the objective. In these processes, the particle size mostly varies between 0.3–2 mm, and the particle density varies between 1.2–4 kg/L.

The societal call for a circular economy (Filho and Sümer, 2015); (Marques et al., 2015) has put pressure on water utilities to change their policies (Ray and Jain, 2011), also in terms of making treatment processes more sustainable. The reuse of waste materials is an example of this transition from a linear to a circular approach. Pellet softening, for instance, is an example of a sustainable process (Beeftink et al., 2020) where full-grown calcium carbonate pellets are crushed and reused as raw material in the process itself (Schetters et al., 2015). The disadvantage, however, is that the processed calcite grains become completely irregularly shaped and show a considerably different hydraulic settling behaviour compared to the generally spherical full-grown calcite pellets. In case of pellet softening processes using fluidisation, the spread in settling velocity can cause the unwanted flushing of smaller grains out of the reactor and the settling of larger grains to the lower region of the reactor, which leads to a fixed bed state. In other processes, like granular activated carbon (GAC) filtration, where bio-based raw materials are getting more attention compared to fossil fuel-based materials, the settling behaviour is important during filterbed backwashing. The physical properties of bio-based grains are often different compared to conventional grain types, which affects the settling behaviour in backwashing processes as well.

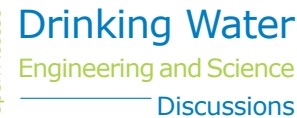

The accurate calculation of the terminal settling velocity of a single particle in water is based on the fluid dynamic drag
coefficient, which is accurately known for spherical particles (Clift et al., 1978). However, accurate prediction models for
settling behaviour of polydisperse highly non-spherical and porous grains applied in filter backwash systems are limited
(Dabrowski et al., 2008); (Hunce et al., 2018). Note that the term 'highly spherical' stands for sphere-shaped particles with a
sphericity ($\Phi \approx 1$), the term 'medium spherical' stands for grains with sphericities ($0.85<\Phi<0.99$) and the term 'lightly non-
spherical' stands for irregularly shaped grains with sphericities ($\Phi<0.95$).

It is important, especially in the field of engineering design and operations for optimal control and optimisation purposes, not
only to accurately predict the drag coefficient and terminal settling velocity, but also to take into consideration the degree of
variation. Aspects such as natural variations in fluid and particle properties, the degree of polydispersity and other factors that
influence the terminal settling velocity were investigated in this work. In this work, we will investigate the amount and the
causes of this spread something which is hugely underexposed in the popular and often cited prediction models presented in
the literature.

### 1.2    Terminal settling and drag coefficient: models from the literature

The literature provides a comprehensive collection of models for the accurate prediction of the terminal settling velocity and
drag coefficient for perfectly round spheres. More recently, advanced drag equations for non-spherical particles have been
proposed, based on geometrical particle properties. With the help of advanced particle image analysis, it is increasingly
possible to determine morphological properties such as sphericity and circularity to predict drag coefficients more accurately.
Nearly all prediction models, based on thorough literature surveys, can be found in the publications listed in Table 1.

***Table 1*** *Publications with overviews of drag coefficient models*

| Spherical particles | Irregularly shaped particles |
|---|---|
| Clift (1978) | Haider–Levenspiel (1989) |
| Concha–Almendra (1979) | Ganser (1993) |
| Brown–Lawler (2003) [1] | Loth (2008) |
| Almedeij (2008) | Hölzer–Sommerfeld (2008) |
| Cheng (2009) | Yang (2015) |
| Barati (2014) | Ouchene (2016) |
| Song (2017) | Bagheri–Bonadonna (2016) |
| Auguste–Magnaudet (2018) | Dioguardi (2018) |
| Goossens (2019) | Breakey (2018) |

[1] Popular drag coefficient prediction models from the literature and a more
detailed explanation of the Brown–Lawler model are included in the
Supplementary Material section (§5)



A very common form of the standard drag coefficient prediction (equation (1)) is an arrangement of groups: laminar $(24/Re_t)$, according to Stokes, transitional $(ARe_t{}^B)$ and turbulent $(C)$, according to Newton (Clift et al., 1978); (Haider and Levenspiel, 1989):

$$C_D = \frac{24}{Re_t}\left(1 + ARe_t{}^B\right) + \frac{C}{1 + \dfrac{D}{Re_t}} \tag{1}$$


with $Re_t$ referring to the (terminal) Reynolds number described in equation (2). Well-known examples for spherical particles are the conventional equation proposed by (Schiller and Naumann, 1933), the equation proposed by (Fair et al., 1971), often applied in water treatment, and the equation proposed by (Brown and Lawler, 2003), covering a wide range of terminal Reynolds numbers. Prediction models for non-spherical particles are also based on the appearance of equation 1. Examples

can be found (Bagheri and Bonadonna, 2016) and (Dioguardi et al., 2018).

### 1.3    Terminal settling velocity calculation

The most common way to calculate the terminal settling velocity $v_t$ is to predict the dimensionless drag coefficient $C_D$ of a single solid sphere in a Newtonian fluid as a function of the Reynolds numbers $Re_t$. The dimensionless particle Reynolds

number under terminal settling conditions is the ratio of the inertial force on the particle to the viscous force with a characteristic length and velocity scale, typically the volume-equivalent particle diameter $d_p$ and terminal velocity:

$$Re_t = \frac{\rho_f d_p v_t}{\eta} \tag{2}$$

To actually predict the steady terminal velocity of a given particle with a projected surface area $A_p$ in the direction of the

gravitational field from these correlations, one needs to consider a force balance in which the drag force balances the difference between buoyancy and weight (Yang, 2003); (Gibilaro et al., 1985); (Clift et al., 1978).

$$\left(\rho_p - \rho_f\right)gV = C_D A_p \frac{1}{2}\rho_f v_t{}^2 \tag{3}$$

For spheres this leads to an analytic *dimensionless* drag coefficient as proposed by (Bird et al., 2007):






$$C_D = \frac{4}{3}\frac{g\,d_p\left(\rho_p - \rho_f\right)}{v_t{}^2 \rho_f} \tag{4}$$

This means that the terminal settling velocity can be calculated by combining equations (1), (2) and (4), assuming that the fluid and particle properties are known. The disadvantage of this set of equations is that the terminal settling velocity must be solved numerically.

The literature also provides empirical equations to predict the terminal settling velocity for specific grains (Concha and Almendra, 1979); (Brown and Lawler, 2003); (Zhiyao et al., 2008).

### 1.4 Aim

The aim of this work was to illustrate the existence of considerable spread in the prediction of terminal settling, a process
which is mostly determined through the prediction of the drag coefficient. This spread becomes relevant as soon as treatment processes must be designed, controlled and optimised. Professionals active in fields where the settling of grains is relevant should be aware of this phenomenon. Merely predicting the drag coefficient and terminal settling velocity based on an estimated average particle diameter, using models derived for perfectly round spheres, is insufficient and likely to be highly inaccurate.

Academic research is predominantly focused on improving the standard drag curve (SDC) for a wide range of Reynolds numbers, from completely laminar to fully turbulent, and researchers regularly present accuracy improvements on a relatively small scale (Almedeij, 2008); (Barati et al., 2014); (Yang et al., 2015); (Whiten and Özer, 2015); (Song et al., 2017) etc. The engineering approach, on the other hand, is focused on higher accuracies mainly for a much smaller operational range. With respect to pellet softening reactors as applied in drinking water treatment processes, the relevant regime is typically
$10 < Re_t < 200$. The present study aimed to improve our understanding of the principles governing the terminal settling velocity of *natural* irregularly shaped particles; the numerical prediction of their terminal settling velocity is much more complex than would be the case for perfectly round particles. To address this, a significant number of terminal settling experiments were carried out and compared with the conventional drag force coefficient equations proposed in the literature (Table 1). Additionally, shape descriptors such as sphericity were measured.

Improved knowledge in this field enables the accurate modelling and optimisation for system and control purposes in automated drinking water treatment processes. This is of value not only for the softening process itself, but also for other processes like the sand-washing processes of seeding material in which dust and undesired materials, such as bacteria, are flushed and released from the process. This is particularly important as unreliable prediction models increase the risk of contamination of the treatment processes, which may adversely affect drinking water quality.




## 2 Materials and methods

### 2.1 Experimental approach

A sequence of different experiments was executed (Table 2). The experimental work started with *old-school* settling experiments with natural, highly irregularly shaped particles and ended with terminal settling experiments using an advanced

calibrated set-up with high-speed cameras. *Old-school* settling entails measuring the vertical fall velocity of grains, visually, in a quiescent fluid using a timer.

The goal of these experiments was to identify the influence of particle size and shape and fluid properties on the terminal settling velocity and settling behaviour.

***Table 2***  *Different types of terminal settling experiments*

| Nr. | Grain type | Study research topic | Shape | Uniformity | Observation |
|---|---|---|---|---|---|
| 1 | Natural and processed | Degree of spread and orientation | Highly non-spherical | Highly polydisperse | Visual |
| 2 | Water softening [1] | Effects of particle growth | Lightly non-spherical | Polydisperse | Visual |
| 3 | Glass beads | Effect of polydispersity | Spherical | Polydisperse | Visual |
| 4 | Glass beads | Agreement prediction models | Highly spherical | Monodisperse | Visual |
| 5 | Glass beads | Wall effects | Highly spherical | Monodisperse | Visual |
| 6 | Glass beads | Individual grain variations | Highly spherical | Monodisperse | Cam [2] |
| 7 | Glass beads | Influence column diameter | Highly spherical | Monodisperse | Visual |
| 8 | Glass beads | Fall length variations | Highly spherical | Monodisperse | Visual |
| 9 | Synthetic | Particle size variations | Spherical | Polydisperse | Visual |
| 10 | Metal balls | Surface roughness | Highly spherical | Monodisperse | Visual |
| 11 | Metal balls | Path trajectories | Highly spherical | Monodisperse | Visual |
| 12 | Calcite pellets and others [5] | Advanced settling | Lightly non-spherical | Polydisperse | 3D cam [3] [4] |

[1] Calcite pellets, crushed calcite and garnet sand
[2] Default traditional camera
[3] 3D trajectory of particle paths using particle tracking velocimetry in a quiescent fluid
[4] Path trajectory videos are shared by (Kramer et al., 2020c)
[5] Metal balls, glass beads and synthetic material


### 2.2 Particle selection

For the terminal settling experiments, opaque and transparent spherical polydisperse and monodisperse glass beads were selected. We also examined drinking water related particles such as calcite pellets and crushed calcite seeding material grains, both of which are applied in drinking water softening. Polydisperse calcite pellets were sieved and separated to acquire more

uniformly dispersed samples. In addition, a wide variety of grains with different densities, sizes and shapes were investigated for their terminal settling velocity and behaviour. The morphological particle properties were obtained with the help of laboratory instruments (Retsch Camsizer XT) and image analysis software (ImageJ).



### 2.3 Experimental set-up

Experimental columns ($D$ = 57 mm) were designed for liquid-solid fluidisation (Kramer et al., 2020a); (Kramer et al., 2020b) and terminal settling experiments, installed at three locations: in Waternet's Weesperkarspel drinking water pilot plant located in Amsterdam, the Netherlands, at the University of Applied Sciences Utrecht, the Netherlands, and at Queen Mary University of London, United Kingdom (Figure 1). Moreover, an experimental column (D = 125 mm) was installed at Waternet and the University of Applied Sciences Utrecht. Finally, an advanced experimental pilot set-up at Delft University of Technology was

used to determine particle 3D trajectories using particle tracking velocimetry in a quiescent fluid (Figure 2).

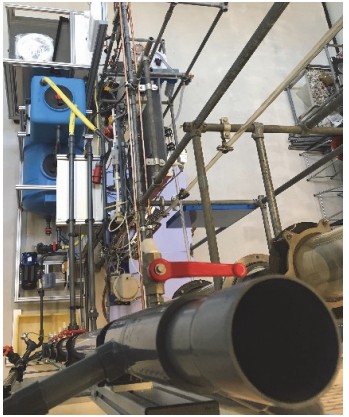

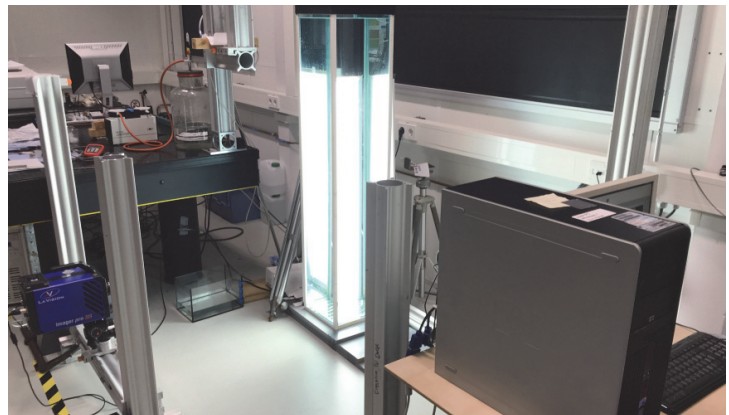

**Figure 1**   *Experimental pilot set-up in Amsterdam, Utrecht and London. (D = 57 mm) with temperature control, column top view*

**Figure 2**   *Experimental pilot set-up at TU Delft (D = 300 mm) to determine particle path trajectories 3D in a quiescent fluid with high-speed cameras*

### 2.4 Procedure

The settling behaviour of single particles was determined for various materials and for different grain sizes. The temperature

was carefully controlled by flowing water through the column of the exact temperature before each experiment and by regularly repeating this process throughout the experiment. Individual particles were dropped at the top of the column. Steady state velocities were reached within one second and before a distance of $L$ = 0.1 m. The condition to be met for steady state velocity is that the particle travels a distance of at least $O\left(100 \cdot d_p\right)$ or greater before the stop clock is switched on. After the steady

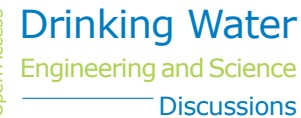

state velocity had been reached, the time required to travel a defined vertical distance ($L$ = 0.50–3.75 m) was measured visually
by the laboratory researcher and the assistant.

### 2.5    Reference data

In addition to the experiments, a large dataset obtained from the literature was examined; this will be discussed in Section 3.5.
The Supplementary Material section includes technical information about experimental set-up devices and flow chart diagram
and procedures (§1), photographic pictures of grains used in water treatment processes (§2) and steady state conditions (§11).

## 3    Results and discussion

### 3.1    Standard drag curve with average values

In total 3,629 new individual terminal settling experiments were executed (Table 2), which marked the starting point of the
spread analysis. Raw data is included in the Supplementary Material section (§17). The results, in accordance with the standard
drag curve approach, are plotted in Figure 3, where experimental results for repeated experiments on sets of the same type and
size of particles have been averaged (symbols). Additionally, the figure includes popular prediction models (lines). The
prediction model equations are included in the Supplementary Material section (§5). A preliminary evaluation (Figure 3)
indicates that the prediction accuracy is reasonably good for most of the grains. Exceptional outliers are wetted-GAC Norit
ROW 0.8 Supra grains (rods), due to particle rotation and their delayed settling behaviour, and the 10 mm glass beads, due to
wall effects.

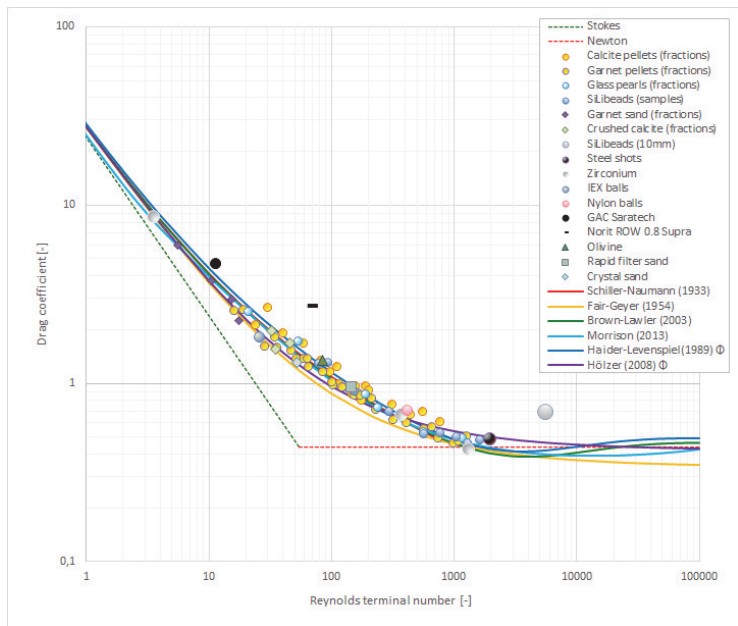

**Figure 3**   *Standard drag curve (SDC) for 3,629 grains, using averaged values over multiple experiments, for 16 types of materials compared with popular prediction models for spherical and non-spherical particles. Φ indicates that the measured sphericity is included in the model. Extended SDC with all examined models is given in the Supplementary Material section (§5, §7)*

Experimental and model results were compared using two statistical error definitions and correlation coefficients; findings are presented in Table 3. To cope with the irregularity of *natural* particles, the measured sphericity was used for models developed for non-spherical particles. With respect to the terminal velocity, the calculated *normalized root mean square* error for the best-known models derived for spherical particles, such as Brown-Lawler and Schiller-Naumann, is in the range of 9-11%.

**Table 3**   *Drag coefficient and terminal settling velocity prediction accuracy for individual terminal settling experiments (N = 3,629). Average relative error (ARE), Normalized root mean square error (NRMSE) and correlation coefficient ($R^2$)*

| Model | ARE ($C_D$) | ARE ($v_t$) | NRMSE ($C_D$) | NRMSE ($v_t$) | $R^2$ ($C_D$) | $R^2$ ($v_t$) |
|---|---|---|---|---|---|---|
| Schiller–Naumann (1933) [1] | 13.0% | 7.0% | 18.4% | 11.3% | 0.91 | 0.93 |
| Fair–Geyer (1954) | 16.7% | 10.1% | 20.2% | 13.1% | 0.89 | 0.96 |
| Clift–Gauvin (1971) | 12.4% | 6.2% | 17.4% | 9.0% | 0.91 | 0.96 |
| Clift (1978) | 12.4% | 6.3% | 17.9% | 9.1% | 0.91 | 0.96 |
| Turton–Levenspiel (1986) | 12.7% | 6.4% | 17.9% | 9.1% | 0.91 | 0.96 |
| Flemmer–Banks (1986) | 13.0% | 6.8% | 18.4% | 9.8% | 0.91 | 0.97 |
| Khan–Richardson (1987) | 12.0% | 6.2% | 17.1% | 9.1% | 0.91 | 0.96 |
| Difelice–Dalavalle (1997) [1] | 22.0% | 9.1% | 27.3% | 10.8% | 0.91 | 0.98 |



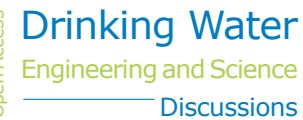

| | | | | | | |
|---|---|---|---|---|---|---|
| Haider–Levenspiel (1989) | 12.6% | 6.4% | 18.0% | 9.1% | 0.91 | 0.97 |
| Brown–Lawler (2003) | 12.1% | 6.2% | 17.1% | 9.0% | 0.91 | 0.96 |
| van Schagen (2008) [1] | 30.2% | 11.8% | 36.7% | 13.7% | 0.90 | 0.97 |
| Cheng (2009) | 12.6% | 6.3% | 18.0% | 9.0% | 0.91 | 0.96 |
| Terfous (2013) | 12.1% | 6.3% | 17.2% | 9.2% | 0.91 | 0.96 |
| Morrison (2013) | 11.8% | 6.2% | 16.8% | 9.1% | 0.91 | 0.96 |
| Barati (2018) | 12.4% | 6.3% | 18.1% | 9.2% | 0.91 | 0.96 |
| Goossens (2019) | 28.5% | 19.8% | 31.6% | 23.3% | 0.86 | 0.96 |
| Haider–Levenspiel (1989) Φ | 14.0% | 6.5% | 20.0% | 8.8% | 0.91 | 0.97 |
| Ganser (1993) Φ | 17.6% | 8.6% | 24.5% | 11.1% | 0.89 | 0.96 |
| Chien (1994) Φ | 17.3% | 9.9% | 22.0% | 13.1% | 0.87 | 0.96 |

[1] Results were rejected when boundary conditions (known limits of applicability) were violated


### 3.2 Drag coefficient and terminal settling velocity prediction versus spread in measured data

Many drag coefficient prediction models found in the literature (Table 1) are based on fits through datasets provided in the literature. Most of the data is based on previous experimental work. In most cases it remains unexplained, and thus unverifiable, whether the literature data represent raw data from single experiments or were processed, for example filtered (by removing

outliers), averaged (using statistics) or corrected (for instance by correcting for wall effects). In the current work we will be explicit about all data processing steps.

To see the amount of variation on an individual particle level, i.e. when no average is calculated, the ratio of the measured to calculated settling velocity (according to Brown–Lawler) was plotted against the calculated settling velocity in Figure 4. To identify statistical outliers, a $3.5\sigma$ bandwidth was added to Figure 4. Of the experimental data, 0.9% can be identified as

outliers. The largest spread is shown for non-spherical particles such as granular activated carbon, olivine, anionic exchange resin (IEX) and garnet grains. In case of garnet sand, outliers can be attributed to the distinctive experimental method of multi-particle settling, i.e. hindered settling (Loeffler, 1953); (Baldock et al., 2004); (Tomkins et al., 2005). As the smallest garnet grains were difficult to detect, multiple grains were settled instead of one single grain. The trends in Figure 4 are prominent, which indicates that individual variability cannot simply be ignored.


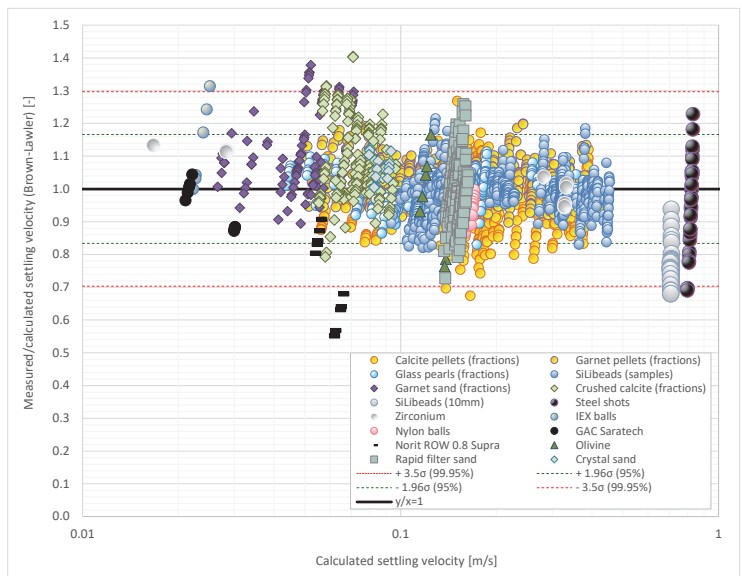

**Figure 4**      *Ratio of measured and calculated terminal settling velocities (Brown–Lawler) against calculated settling velocity. Statistical probability estimation 95% ($\mu \pm 1.96\,\sigma$) plot and the ($\mu \pm 3.5\,\sigma$) to show the outliers (0.9%) 32 of 3,629 experimental values. A similar graph for the drag coefficient is given in the Supplementary Material section (§9).*

### 3.3      Uncertainty analysis

To better estimate the consequences of spread and accordingly to be able to compensate this in full-scale operational processes, it is important to know which parameters cause the spread in drag and observed settling velocity. We started with an uncertainty
analysis to estimate the degree of deviation in variables caused by the following uncertainties in measured parameters (Table 4), to be able to add error bars to the standard drag curves. The estimates of uncertainty in $C_D$ and $Re_t$ as well as in $\rho_p$ and $v_t$ were calculated according to the propagation of errors method (Ku, 1966).

**Table 4**  *Decisive variable and parameter investigation*

| Variables | Parameters |
|---|---|
| $C_D$ is determined by: | $g, d_p, \rho_p, \rho_f, v_t$ |
| $Re_t$ is determined by: | $d_p, \rho_f, v_t, \eta$ |
| Direct measurements: | Particle properties: $d_p, \rho_p, \Phi$ |
| | Fluid properties: $\rho_f, T$ |
| | Experimental: $g, D, L, t$ |



Figure 5 shows typical results for the uncertainty in $C_D$ versus $Re_t$ for 16 selected particle types, expressed with error bars. Results for all other particle types as well as detailed derivations of the contribution to the errors can be found in the Supplementary Material section (§6, §7).

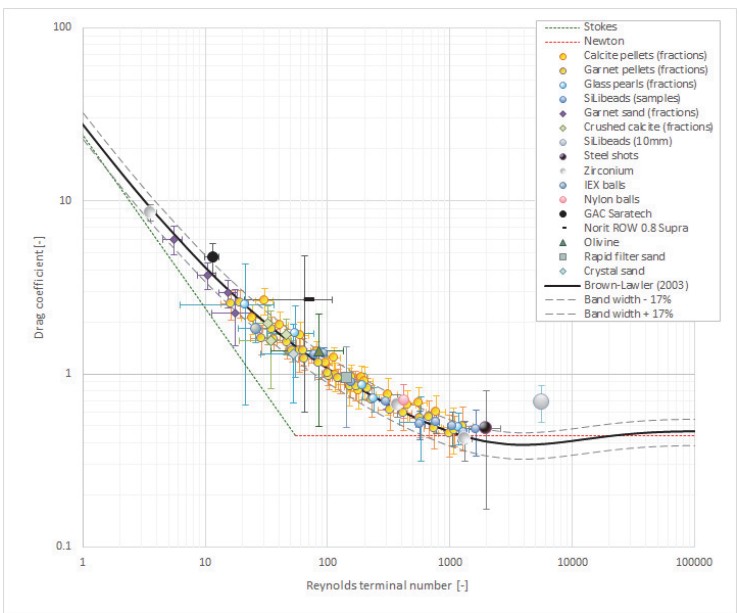

**Figure 5**     *Standard drag curve with error bars based on uncertainty analyses for 16 types of materials compared with the Brown–Lawler prediction model. A bandwidth of 35% is added, based on a summarised propagated effect of errors on the uncertainty of the experimental measurements. Error bars for specific particle types and specific research aims (Table 2) can be found in the Supplementary Material section (§6, §7).*

**3.3.1**     **Natural and processed highly non-spherical polydisperse particles**

Natural irregularly shaped particles often used in water treatment processes, such as olivine, calcite, GAC grains and several sand types, cause the largest degree of spread in the standard drag curve. $C_D$ values for GAC grains are higher compared to the calculated value according (Brown and Lawler, 2003). However, spherical GAC grains show a slightly lower spread, with an error $\delta C_D \approx 1$. The error for the rod-shaped GAC grains is considerable larger due to the combination of a large particle size

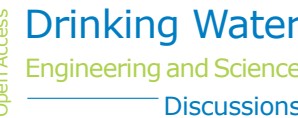

distribution (PSD), non-spherical shape, particle orientation and particle porosity. During the experiments, it was visually observed that the GAC rods tended to settle horizontally. Additionally, they showed wobbling and zigzag behaviour.

The settling behaviour in terms of drag for olivine, crystal sand, garnet sand and rapid filter sand is less erratic. It is notable that particularly for rapid filter sand grains the error in $Re_t$ is large compared to the error in $C_D$ (Haider and Levenspiel, 1989). This is mainly due to a large PSD, i.e. grains were originally mined and not sieved in advance. The non-spherical particle

properties are less decisive. The observed spread for other natural grains is similar. However, for grains smaller than 0.5 mm, detecting the settling velocity became more complex and challenging.

For crushed calcite pellets, the error in $C_D$ mainly results from the grains' irregular shape caused by their processing, i.e. grinding (Schetters et al., 2015). As the grains were sieved, the PSD is less wide.

The SDC curve for natural and processed highly non-spherical polydisperse particles is given in the Supplementary Material

section (§7.1).

### 3.3.2    Medium non-spherical polydisperse particles used in water softening

Calcite pellets were extracted from the water softening reactor, dried and fractionated using calibrated sieves. Detailed morphological properties such as sphericity and circularity were also measured; these are included in the Supplementary

Material section (§2). The extra information was used in the prediction models. The prediction accuracy for $C_D$ was calculated for models derived for spherical particles and for models derived for non-spherical particles. Table 5 presents the accuracy, where the symbol Φ stands for including the particles' morphological properties. No prediction model can predict the drag coefficient with an error level below 10%. The 'best' results are obtained by the classical Haider–Levenspiel model and, with a slightly lower score, the Brown–Lawler model.


**Table 5**    *Drag coefficient prediction accuracy for calcite pellets (0.36<$d_p$<2.8 mm) based on individual terminal settling experiments (N = 1,163)*

| Model | ARE ($C_D$) | NRMSE ($C_D$) | $R^2$ ($C_D$) |
|---|---|---|---|
| Fair–Geyer (1954) | 15.4% | 17.2% | 0.91 |
| Brown–Lawler (2003) | 12.1% | 13.5% | 0.90 |
| Morrison (2013) | 12.7% | 14.5% | 0.91 |
| Goossens (2019) | 37.1% | 38.1% | 0.88 |
| Haider–Levenspiel (1989) Φ | 10.0% | 11.2% | 0.91 |
| Ganser (1993) Φ | 13.1% | 15.7% | 0.91 |
| Chien (1994) Φ | 19.6% | 21.0% | 0.91 |
| Hölzer (2008) Φ | 21.0% | 24.5% | 0.87 |
| Bagheri (2016) Φ | 15.2% | 19.8% | 0.85 |
| Dioguardi (2017) Φ | 25.7% | 28.6% | 0.92 |

Figure 6 presents the average $C_D$ values for calcite pellets where, from a visual perspective, the dots show a reasonable fit with the majority of the models. The error bars clearly show that the variation in the measured data constrains the prediction

accuracy. Detailed morphological data of calcite pellets and crushed calcite and the standard drag curve for natural and

processed highly non-spherical polydisperse particles are given in the Supplementary Material section (§2, §7.2).

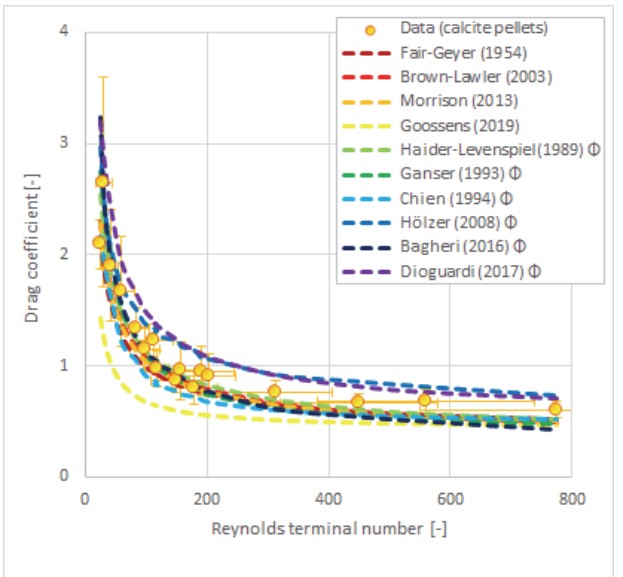

**Figure 6**    *Drag coefficient prediction (average values on lin–lin scale) accuracy for calcite pellets (0.36<$d_p$<2.8 mm) including (measured) error bars, based on individual terminal settling experiments (N = 1,163). Φ indicates that the measured sphericity is included in the model.*

### 3.3.3    Highly spherical polydisperse and monodisperse glass beads

In the literature, glass beads are popular and frequently used for model calibration and validation purposes. In this work, 288
individual spherical glass pearls were settled. The $C_D$ values show reasonable agreement with the Brown–Lawler curve. Data
spread is caused by polydispersity (UC >1), albeit less pronounced than for calcite pellets. A whole sequence of highly
spherical (Φ →1) monodisperse (UC →1) glass beads (N = 911) was studied in terms of their settling behaviour. For these
particles, diagonal trends in the SDC plots were noticeable, despite the fact that the average $C_D$ coincides fairly well with the

Brown–Lawler curve. These trends are related to the way the estimated drag coefficient depends on the measured settling
velocity (equation (4)) and have been observed by (Veldhuis et al., 2009) and (Raaghav, 2019). The slope in the standard drag



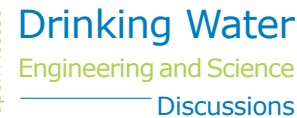

curve equals approximately -2, corresponding to $C_D'/\overline{C_D} \sim -2\,v_t'/\overline{v}_t$. A mathematical basis for this trend is explained in the Supplementary Material section with the help of a simple scaling analysis (§10) and the SDC curve (§7.3).

**3.3.4    Repetitive experiment with highly spherical monodisperse single glass beads**

To eliminate the human factor, one sequence was executed where one and the same single glass bead ($d_p$ = 3 mm) was used 30 times. The start and end times were filmed, and hence the error in $\delta t$ and $\delta L$ was negligible. We observed that the spread in $C_D$ decreased when human error was excluded. Nevertheless, similar to spherical monodisperse glass beads, the previously observed trend (slope -2) was observed. The SDC curves for highly spherical polydisperse and monodisperse single glass

beads are given in the Supplementary Material section (§7.4).

**3.3.5    Highly spherical monodisperse glass beads and wall effects**

Additional to the highly spherical monodisperse glass beads, larger glass beads ($d_p$ = 10 mm) were tested in a small cylindrical column ($D$ = 57 mm). In this particular case, wall effects evidently played a role in the retardation of the terminal settling

velocity. In addition, it became apparent during the experiments that the glass beads tended to move to the wall, followed by a prominent zigzag movement due to side drifting motions caused by a high Galileo number (Zhou and Dušek, 2015). The Galileo number is expressed in equation 5:

$$Ga = \sqrt{\frac{g\,d_p^{\,3}\rho_f(\rho_p - \rho_f)}{\eta^2}} \tag{5}$$

In the standard drag curve, $C_D$ is higher compared to the Brown–Lawler curve, but this can be attributed to wall effects and non-vertical settling trajectories. Wall effect correction equations given by (di Felice and Gibilaro, 2004); (Gibilaro et al., 1985) and (Chhabra et al., 2003), often empirically based, could not compensate for these non-ideal phenomena and circumstances. The SDC curve and a video fragment illustrating the wall effects for a highly spherical monodisperse glass bead are given in the Supplementary Material section (§7.5).


**3.3.6    Highly spherical monodisperse glass beads in different columns and with different fall lengths**

To explore the influence of the column diameter, the same experiments were executed in two columns with different sizes ($D$ = 57 mm and $D$ = 125 mm) for three different glass bead sizes (1.5, 2.5 and 3.5 mm). The successive experiments aimed to determine whether the fall length plays a role. An important aspect here is that the settling velocity was measured merely in a

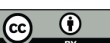



steady state situation. Based on the results and figures given in the Supplementary Material section (§7.6, §7.7), no distinction
can be made.

### 3.3.7    Highly spherical polydisperse synthetic particles

Spherical polydisperse particles with a low particle to fluid density ratio ($\bar{\rho}$ = 1.4) have similar settling behaviour. Nylon balls
and IEX resin balls are spherical and have a relatively high uniformity coefficient. IEX balls are more polydisperse compared
to nylon balls and show more spread in $C_D$. The SDC curve for highly spherical polydisperse synthetic particles is given in the
Supplementary Material section (§7.8).

### 3.3.8    Highly spherical monodisperse metal balls

The outside layer of the examined zirconium balls is $ZrO_2$, so the surface is not smooth. To investigate if this affects the drag,
we tested highly spherical, monodisperse zirconium balls with three different sizes (0.1, 1.0 and 2.0 mm). The individual
measured drag coincides well with the Brown–Lawler curve. Generally speaking, surface roughness can cause the boundary
layer to become turbulent and the wake region behind the sphere to become considerably narrower than if it were laminar,
which results in a considerable drop in pressure drag with a slight increase in friction drag (Munson et al., 2009); (Loth, 2008);
(Bagheri and Bonadonna, 2016). Nevertheless, the influence of particle surface roughness on the drag coefficient for Reynolds
($Re_t$<40,000) can be neglected. The range for Reynolds in this work for all experimental data is $1.2 < Re_t < 7,500$; surface
roughness effects were not found and therefore further neglected.

Additionally, the settling behaviour of highly spherical and monodisperse metal balls ($d_p = 3\ mm$) with a high particle to
fluid density ratio ($\bar{\rho} = 8$) was studied in a cylindrical column ($D = 125$ mm). Based on the average measured $C_D$, the
estimated drag was 7% smaller than the Newton constant drag ($C_D = 0.44$), but 20% above the Brown–Lawler predicted value
in this particular range of Reynolds number. The measured drag had a small calculated spread ($C_D = 0.41 \pm 0.01$). The
experiments, however, prove the existence of a substantial discrepancy at the individual level. We observed four different
settling behaviours and path trajectories (Table 6). An average drag coefficient $C_D = 0.48$ +/- 0.17 was determined for $N = 35$
individual measurements. Nonetheless, a lower $C_D = 0.34$ +/- 0.04 was determined for metal balls with a vertical path
trajectory, and $C_D = 0.44$ +/- 0.04 for particles which tended to move to the wall but did not touch it. For particles that came
into contact with the wall of the tube, a significantly higher $C_D = 0.68$ +/- 0.14 was found. In this particular case, the wall
effect, causing retardation of the settling velocity due to water displacement, is another factor to be considered. Here, the wall
effect implies wall contact interactions. Similar to what was found for 10 mm glass beads, we observed a vertical bouncing
effect on the wall caused by a chaotic zigzag fall trajectory (Zhou and Dušek, 2015).
The SDC curves for highly spherical monodisperse metal balls are given in the Supplementary Material section (§7.9).



***Table 6*** Path trajectories of 3.0 mm steel shots

| Wall interaction | Description trajectory |
|---|---|
| None | No wall effects, a straight vertical fall path |
| Minor | Tends to move to the wall but does not touch it |
| Moderate | Moves towards the wall and touches it |
| Considerable | Touches the wall from the beginning |

### 3.4    Path trajectories

The path trajectories of fractionated calcite pellets (1.0 mm – 2.8 mm) were recorded using an advanced experimental set-up.

Figure 7 and Figure 8 present the path trajectories of single calcite pellets, demonstrating the non-linear fall trajectory of grains.

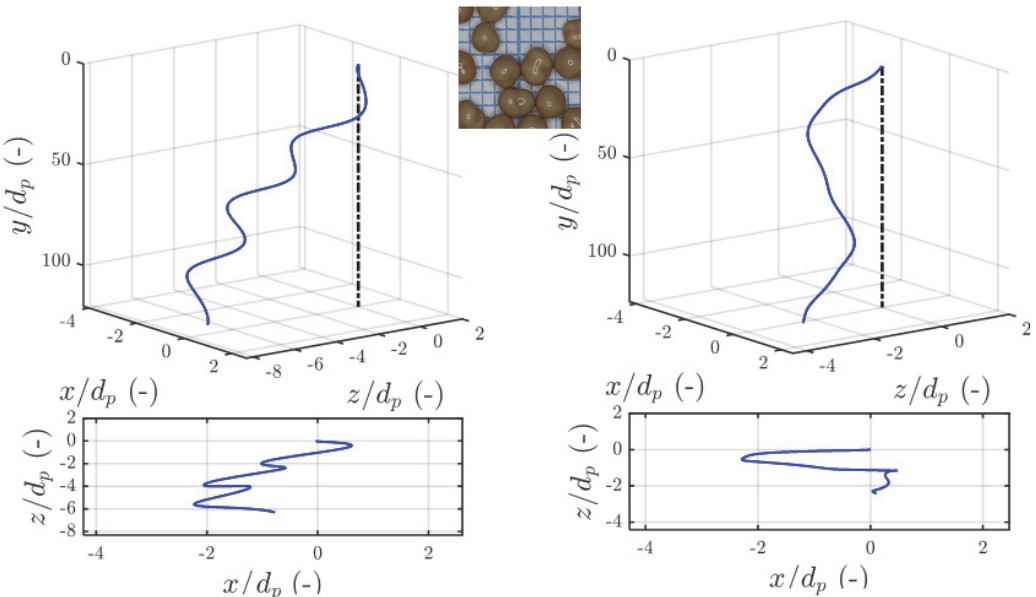

**Figure 7**  *Calcite pellets: 2.36<$d_p$<2.8 mm, T = 20 ℃, $C_D$ = 0.55, $\bar{\rho}$ = 2.7, $Re_t$ = 809, $Ga$ = 522, $v_t$ = 0.32, $v_{t,BL}$ = 0.34, angle = 2.8°, −%$v_t$ = 7%, path: CH*

**Figure 8**  *Calcite pellets: 2.36<$d_p$<2.8, T = 20 ℃, $C_D$ = 0.57, $\bar{\rho}$ = 2.7, $Re_t$ = 800, $Ga$ = 522, $v_t$ = 0.32, $v_{t,BL}$ = 0.34, angle = 2.1°, −%$v_t$ = 8%, path: CH*



Chaotic paths of freely falling and ascending spheres, path instabilities and transitions in Newtonian fluid have been discussed by many (Jenny et al., 2004); (Veldhuis and Biesheuvel, 2007); (Horowitz and Williamson, 2010); (Zhou and Dušek, 2015); (Auguste and Magnaudet, 2018); (Riazi and Türker, 2019) and experimentally proven by (Raaghav, 2019). To investigate the path trajectories expected for our particles, we investigated the state diagram (Zhou and Dušek, 2015) of Galileo number $Ga$ versus density ratio $\rho_p/\rho_f$ in Figure 9, magnified in Figure 10. The state diagram contains different areas with typical settling behaviours. Several areas overlap, which means that different trajectories might occur. The regime map proposed by Zhou and Dušek is derived for perfect spheres. Path trajectories for calcite pellets will not follow the regime map completely, due to their less regular shape, as the regimes are sensitive to the anisotropic nature of the particles. The measured sphericities of calcite pellets, given in the Supplementary Material section (§2), however, have values larger than Φ>0.9, so we expect a qualitatively similar path trajectory behaviour.

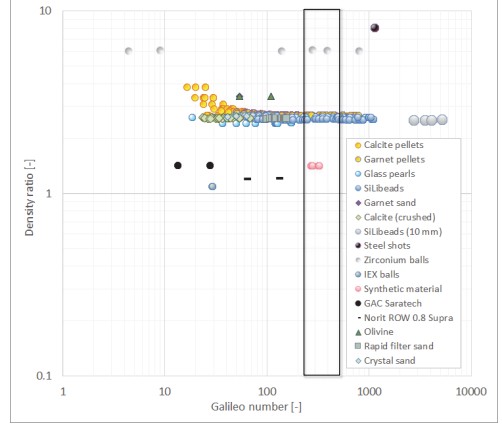

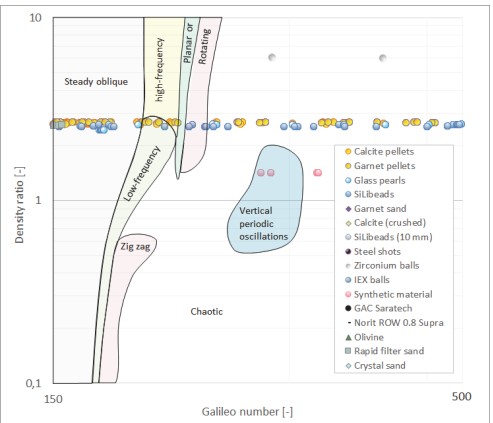

***Figure 9*** *State diagram. Galileo number $Ga$ versus specific gravity number $\bar{\rho}$ with examined particles*

***Figure 10*** *Path trajectory regime plot according (Zhou and Dušek, 2015; Raaghav, 2019) (zoomed area)*

As shown in Figure 10, about 3/4 of all examined grains belong to the steady oblique regime and 1/4 to the three-dimensional chaotic regime. Calcite pellets show a similar pattern: 4/5 steady oblique and 1/5 chaotic. Glass beads: 1/2 steady oblique and 1/2 chaotic. Synthetic material and metal balls belong almost completely to the chaotic regime.

Individual path trajectory behaviour of the examined calcite pellets and of other particles are given in the Supplementary Material section (§8). Path trajectory videos are shared by (Kramer et al., 2020c).






### 3.5    Data from the literature

In the literature, raw and processed settling data is available for research purposes. The dataset generated by (Brown and Lawler, 2003) is a composition of previous research experiments on spherical particles ($N = 480$). Other researchers (Wu et al., 2006); (Almedeij, 2008); (Cheng, 2009); (Dioguardi and Mele, 2015); (Song et al., 2017); (Dioguardi et al., 2018);

(Breakey et al., 2018) shared data for both spherical and non-spherical particles. Based on literature data, $C_D$ versus $Re_t$ for $N$ = 3,655 data points is plotted in Figure 11. Figure 12 shows a smaller area, focusing on covering the relevant regime for water treatment, where the data spread reconfirms the apparent spread and deviations also found in our work. A data spread of, for instance, +/- 50% in $C_D$ means a factor 0.8–1.4 in $v_t$. The consequences for a sand wash installation, for example, is an error in $v_t$ of +/- 20%, which could raise the question whether this is sufficiently accurate and suitable for process control. Drag

coefficient prediction accuracy, similar to the data in Table 3 and data from the literature, is given in the Supplementary Material section (§5.7, §17).

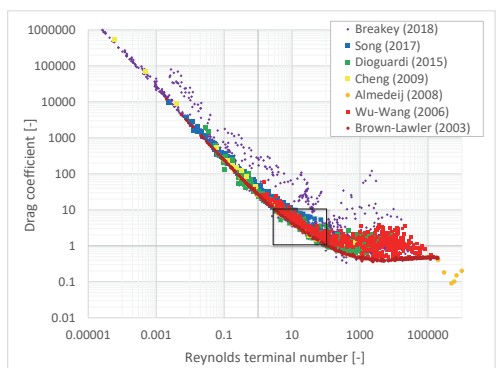

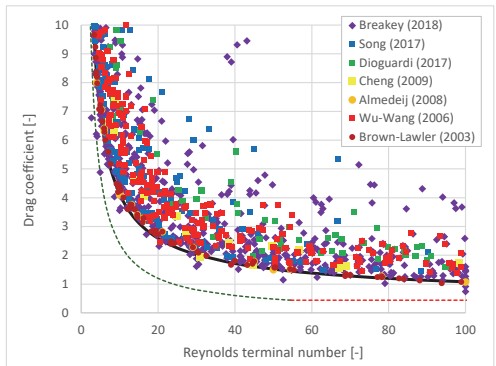

***Figure 11***    *SDC (log–log) data from literature sources.*

***Figure 12***    *SDC (lin–lin) data from literature sources, zoomed in on an area that is important for drinking water treatment. Solid lines represent the Brown–Lawler drag model and dashed lines the Stokes and Newton drag models, respectively.*

### 3.6    Propagated effect of parameter uncertainties on terminal settling

Figure 13 shows the influence of the uncertainty in various parameters on uncertainty in the settling velocity $v_t$. The summarised propagated effect of errors on the uncertainty of the experimental measurements are 35% for the terminal settling velocity and 56% for the terminal Reynolds number. The graphically summarised propagated effect of errors for $C_D$ and $Re_t$

are presented in the Supplementary Material section (§6). The figure shows that some causes, like variation in gravity, surface roughness and linear expansion due to temperature changes, can be neglected.

Uncertainties in the fluid density and viscosity as well as in the estimated (human) error of measurements have a relatively minor effect on the error in $v_t$. For instance, the error in $v_t$ resulting from the human error in measurements is estimated at 1.3%, based on human response time inaccuracies. Depending on the tube and particle dimensions, also wall effects, leading to retardation of the settling velocity, can be ignored, certainly in full-scale systems. Based on the wall effect equation proposed by (Arsenijević et al., 2010), which has gained wide acceptance in the literature, the error on $v_t$ is estimated to be 2.6% for all

measurements made in this study. Further details and an explanation with respect to wall effects and error analysis can be found in the Supplementary Material section (§13).

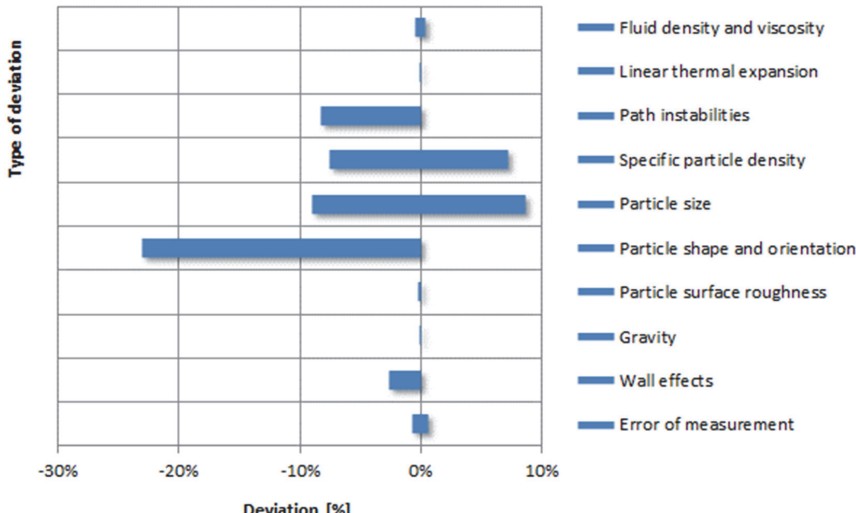

**Figure 13**    *Summarised propagated effect for the terminal settling velocity resulting from different causes*

Figure 13 shows that the vast majority of the spread is caused by variations in specific particle density, particle size and shape,

particle orientation and path instabilities. The error caused by natural variations in particle density *combined* with the relative error of experimentally measured particle density in the laboratory was approximately 7%. The error in $v_t$ was calculated at 14.8%.

Regarding particle size, in this work pellets were sieved to produce more monodisperse particle samples. On the assumption that spheres are round and pass the sieves, the variation in size ($\Delta d_p$) is 19.0%, but this depends on the type of sieve used. The

variation in diameter had a considerable effect (17.7%) on spread for $v_t$. A special case is included in the Supplementary



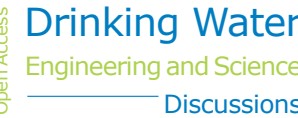
Material section (§12), based on the assumption that irregularly shaped particles behave like spheroids. It is illustrated how particles pass a sieve and rotate and settle in comparison with a particle with another projected surface.

The literature shows that the effect of particle orientation on the drag coefficient depends on particle shape, (Abraham, 1970); (Bird et al., 2007); (Loth, 2008); (Bagheri and Bonadonna, 2016). As the particle shape becomes less spherical, the effect of

particle orientation becomes more significant due to the increase of the ratio between maximum and minimum projected areas. In the Stokes regime, the particles do not have a preferred orientation and for a statistically representative run of experiments they can adopt any random orientation.

An easy preliminary approach for non-spherical shapes can be adopted through the sphericity Φ, which is frequently used in drinking water treatment processes to correct for irregular particles (Yang, 2003). The sphericity of a particle is the ratio of the

surface area of a sphere with the same volume as the given particle to the surface area of the particle. In the case of sand grains, the drag coefficient increases from 1.2 to 1.7 when the sphericity decreases from Φ = 1.0 to Φ = 0.7, which corresponds to a 20% increase of $C_D$ (US-IACWR, 1957). When the sphericity is decreased stepwise by 10%, the terminal settling velocity decreases linearly by 10% while, in contrast, the drag coefficient increases almost twice as much. For sand, (Ðuriš et al., 2013) selected a reasonable sphericity Φ = 0.76. According to (Yang, 2003), the sphericity varies between Φ = 0.66 for sharp sand

and 0.86 for round sand, which agrees well with Geldarts observation for measured settling velocity (Geldart, 1990). A sphericity of 0.66 results in a 23.0% decrease of $v_t$ and a 28.6% increase of $C_D$.

(Albright, 2009) showed that for cylindrical particles towards the laminar regime ($Re_t$ < 50) the drag coefficient is lower compared to round spheres. However, this coefficient is higher for more turbulent regimes where $Re_t$ > 50. (Dharmarajah, 1982) reported that under creeping conditions all orientations are stable ($Re_t/\Phi$ < 0.1) and that in the transitional regime (0.1

< $Re_t/\Phi$ < 200) particles are stable since they tend to orient themselves with the largest cross-section in the three mutually perpendicular planes of symmetry in a position normal to the direction of motion. Under more turbulent conditions (200 < $Re_t/\Phi$ < 500), the orientation of settling is less predictable: examples include wobbling and rotation. For the inertial regime ($Re_t/\Phi$ > 500), the particles' rotation about their axis is frequently coupled with spiral translations. (Haider and Levenspiel, 1989) demonstrated in the drag–Reynolds terminal diagram that for irregular particles with increasing non-sphericity, the drag

coefficients also increase considerably: this can rise by as much as 500%. This demonstrates that for higher Reynolds numbers irregularity becomes increasingly important.

### 3.7 Consequences of uncertainty in settling velocity for water treatment processes

The discussion on how to measure the terminal settling velocity of a single particle, or multiple particles, is extremely relevant.

What is the most representative for a full-scale system? Not a single particle. Hence, it is important to discuss how single particle measurements can be extrapolated to information relevant for the full-scale system. Often the settling velocity is expressed as a fraction of the terminal settling velocity. For instance, in their famous article (Richardson and Zaki, 1954) the



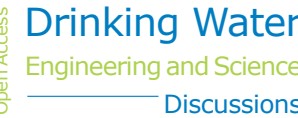

settling velocity of multiple particles for a voidage extrapolated to 1 equals the apparent free-falling settling velocity of a single
particle at infinite dilution, i.e. the terminal settling velocity $v_t$. Many water treatment processes like pellet softening and filter
backwashing operate at a voidage in the vicinity of the incipient state (Kramer et al., 2019). Therefore, a large uncertainty in
$v_t$ has a considerable effect on the voidage prediction. In this work, we have explicitly shown the causes of uncertainty in $v_t$.
There is no model, for the prediction accuracy for terminal settling velocity and drag coefficient, that covers the wide range of
differences in particle properties with a low prediction inaccuracy (<1%). The prediction accuracy for models derived for non-
spherical particles (Haider and Levenspiel, 1989); (Ganser, 1993); (Hölzer and Sommerfeld, 2008); (Ouchene et al., 2016),
using the sphericity as a shape descriptor, is not significantly improved for drinking water related granules.

## 4  Conclusions

Based on measured average terminal settling velocities, drag prediction models like Brown–Lawler were found to agree
reasonably well with experimental observations. However, individual terminal settling velocities showed a considerable
amount of spread around the average value. In general, particle size and shape variations as well as chaotic path trajectories
during settling are the most decisive reasons why the spread in individual terminal settling velocities occurs. In this work we
observed two kinds of wall effects. Besides their decreased settling velocity, the aspect that is the most frequently discussed
in the literature, particles also show variations in path trajectories where they touch the vessel wall, thus leading to a reduced
velocity.

While the majority of the predictive correlations lie within a bandwidth of 6% between each other, the summarised propagated
effect of errors on the uncertainty of the experimental measurements is 34% for $v_t$, 35% for $C_D$ and 56% for $Re_t$. The data
obtained from literature sources also show a considerable degree of spread in $C_D$. The terminal settling velocities determined
with an advanced experimental set-up were compared with *old-school* velocities measured by eye and stopwatch. The average
relative error between the two methods was only 4% +/- 3%, so this cannot explain the observed large spread in individual
measurements. Simple models such as $C_D = \frac{24}{Re_t} + 0.44$ (Goossens, 2019) have a relatively low prediction accuracy, based on
the data acquired. Nevertheless, one should take into consideration the existing data spread around the average $C_D$ when other
models are used with apparently higher prediction accuracies. In other words: more complex expressions do not automatically
entail higher accuracy.

Our results have important implications when drinking water treatment processes are optimised or designed, for instance with
a new type of grain with specific morphological, density or other particle properties. It is important to take notice of the spread
in settling velocities. The considerable degree of spread in terminal settling velocities could result in less optimal process states
and lower efficiency in the use of raw materials and should therefore be taken into account in the design, operation and
optimisation of water treatment processes.

Finally, the prediction accuracy for terminal settling velocity and drag coefficient should be improved, in particular for non-
spherical particles.




In conclusion, to answer our main question whether terminal settling velocity and drag of natural particles in water can ever be predicted accurately, we have to say 'yes, it is possible', at least for spherical particles and using a model such as Brown–Lawler. The answer is 'possibly yes' for non-spherical particles, albeit only when more morphological properties are included besides (equivalent) particle diameter, circularity and sphericity. During the past decades, novel work has been published on

the topic of terminal settling, Nevertheless, some puzzles remain unsolved. The prediction accuracy can be improved by means of new advanced research, to be carried out in academia as well as in industry.

**Nomenclature**

Subscripts, superscripts and abbreviations can be found in the Supplementary Material section (§15).


**Symbols**

|  |  |  |
|---|---|---|
| $a, b, c$ | Radius of spheroids | [m] |
| $A, B, C, D$ | Coefficients | [-] |
| $Ar$ | Archimedes number | [-] |
| 475   $A_p$ | Particle projected area | [m$^2$] |
| $A_s$ | Area of spherical particle | [m$^2$] |
| $c_i$ | Coefficients | [-] |
| $C_D$ | Fluid dynamic drag coefficient | [-] |
| $\overline{C_D}$ | Average drag coefficient | [-] |
| 480   $C_D'$ | Error / uncertainty introduced in drag coefficient | [-] |
| $D$ | Inner column or cylinder vessel diameter | [m] |
| $d_g$ | Average seeding material diameter | [m] |
| $d_i$ | Effective size of a sample where $i$ percentage of particles is smaller than the particular size | [m] |
| $d_p$ | Effective or average or particle equivalent diameter | [m] |
| 485   $d_{s,i}$ | Sieve mesh diameter | [m] |
| $error$ | 1.96 times standard spread |  |
| $E_{H,50}$ | Ellipsoid height (cumulative 50% point) | [m] |
| $E_{L,50}$ | Ellipsoid length (cumulative 50% point) | [m] |
| $E_{W,50}$ | Ellipsoid width (cumulative 50% point) | [m] |



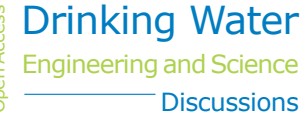

| 490 | $F_b$ | Buoyancy force exerted by a fluid that opposes the weight of an immersed object | [N] |
| | $F_D$ | Drag or frictional force of a spherical particle during terminal settling | [N] |
| | $F_g$ | Force by the gravitational field | [N] |
| | $F_p$ | Net force exerting on spherical particle under terminal settling conditions | [N] |
| | $Ga$ | Galileo number | [-] |
| 495 | $g$ | Local gravitational field of earth equivalent to the free-fall acceleration | [m/s²] |
| | $k$ | Wall effects correction multiplier | [-] |
| | $m$ | Particle mass | [kg] |
| | $n$ | Richardson–Zaki coefficient, expansion index | [-] |
| | $N$ | Total number of particles / total number of experiments | [#] |
| 500 | $Re$ | Reynolds number, ratio of inertial forces to viscous forces within a fluid | [-] |
| | $Re_t$ | Reynolds terminal number | [-] |
| | $Re_p$ | Reynolds particle number | [-] |
| | $Symm$ | Symmetry, the distances between the centre of area to the particle projection borders | [-] |
| | $UC$ | Non-uniformity coefficient $d_{60}/d_{10}$ | [-] |
| 505 | $\bar{v}_t$ | Average terminal settling velocity | [m/s] |
| | $v_t{}'$ | Error / uncertainty introduced in velocity | [m/s] |
| | $v_i$ | Apparent free-falling settling velocity of a particle in an infinite dilution | [m/s] |
| | $v_s$ | Linear superficial velocity or empty tube fluidisation velocity | [m/s] |
| | $v_t$ | Terminal particle settling velocity | [m/s] |
| 510 | $v_{t,BL}$ | Terminal settling velocity according Brown-Lawler | [m/s] |
| | $T$ | Temperature | [°C] |
| | $V$ | Volume | [m³] |
| | $V_p$ | Volume of spherical particle | [m³] |
| | $x$ | Average particle diameter between top and bottom sieves | [m] |
| 515 | | | |

**Greek symbols**

| | $\alpha$ | Linear heat expansion coefficient | [m/mK] |
| | $\delta$ | Uncertainty | |
| | $\varepsilon$ | Voidage of the system | [m³/m³] |
| 520 | $\eta$ | Dynamic fluid viscosity | [kg/(ms)] |
| | $\lambda$ | Ratio between average particle grain diameter and inner column diameter | [-] |



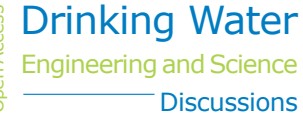

|  | | | |
|---|---|---|---|
| $\mu$ | Statistical mean | | |
| $\bar{\rho}$ | Specific gravity number, particle to fluid density ratio $(\rho_p/\rho_f)$ | | [-] |
| $\rho_c$ | Density of calcium carbonate | | [kg/m³] |
| $\rho_f$ | Fluid density | | [kg/m³] |
| $\rho_g$ | Seeding material density | | [kg/m³] |
| $\rho_p$ | Particle density | | [kg/m³] |
| $\upsilon_T$ | Kinematic fluid viscosity | | [m²/s] |
| $\sigma$ | Standard spread | | |
| $\phi_s$ | Shape of diameter correction factor | | [-] |
| $\Phi$ | Sphericity: ratio between surface area of the volume equivalent sphere and considered particle $\frac{\pi^{\frac{1}{3}}(6V_p)^{\frac{2}{3}}}{A_s}$ | | [-] |
| $\Phi_\perp$ | Crosswise sphericity | | [-] |
| $\Phi_\parallel$ | Lengthwise sphericity | | [-] |
| $\Psi$ | Particle shape descriptor | | [-] |
| $\Xi$ | Circularity calculated from the perimeter P and area A of the particle projection $\sqrt{\frac{4\pi A_p}{P^2}}$ | | [-] |

**Acknowledgements**

This research is part of the project "Hydraulic modelling of liquid-solid fluidisation in drinking water treatment processes" carried out by Waternet (the water utility of Amsterdam and surroundings), Delft University of Technology and HU University of Applied Sciences Utrecht. Financial support came from Waternet's Drinking Water Production Department. For our simulation, we used Symbolic regression Software Eureqa. We thank Nutonian for allowing us to use their software.

We acknowledge and thank our students from Delft University of Technology, HU University of Applied Sciences Utrecht and Queen Mary University of London, and in particular Victor Shao and Cas van Schaik for the precise execution of many laboratory and pilot plant experiments. We also thank Desmond Lawler, Professor at the University of Texas and Dr. Reza Barati of the Tarbiat Modares University for sharing the original drag and terminal settling data. Finally, we thank Dr. Wim-Paul Breugem of Delft University of Technology for making the advanced laboratory equipment available to track the 3D motion of single settling grains and to capture their path instabilities.

This research project did not receive any specific grant from funding agencies in the public, commercial or not-for-profit sectors.



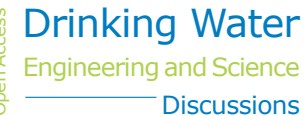

**Declaration of Competing Interest**

The authors declare that they have no known conflicts of interests or personal relationships that could have appeared to influence the work reported in this article.

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
