# Peer review of "Can terminal settling velocity and drag of natural particles in water ever be predicted accurately?"

_Drinking Water Engineering and Science, 2020_

## Referee Comment (RC1) · Anonymous Referee #1 · 13 Nov 2020

This paper is a very detailed discussion of the spread of 3,629 terminal settling experiments and of related literature data. Scientific sustained arguments leads to the conclusion that new advanced research is needed to improve the prediction accuracy for settling velocity, drag coefficient and terminal Reynolds number of non-spherical particles such as drinking water related particles. The quality of the paper has to be upgraded by the following minor revisions: - Line 87: according to. . .. Replace Newton by Clift . - Line 116: add: Nian-Sheng Cheng 1997 Terfous et al. 2013. Goossens 2020. - Line 190-191: „„standard drag curve„, ADD (Lapple-Shepherd 1940) - Line 290: this definition of the Galileo number has to be repeated in the list of symbols . comment: this definition is peculiar as in a standard way the square root is omitted. -

Line 325: replace "estimated" by "experimental". - List of symbols: are not used in the paper and have to be removed of the list: A,b,c, Ar As ct di dp dsi Eh50 E1.50 Ew50 Fb Fd Fg Fp k Symm UC x Greek: likewise. . ... - List of symbols: Are to be defined: Cd =definition of eq. (4) Cd' _ Cd dp is volume-equivalent particle diameter Ga = definition of eq. (5) Ret = definition of eq. (2) - References : ADD: Nian-Sheng Cheng, J. Hydraulic Eng. February 1997,149-152 Walter R.A. Goossens, Powder Technology 362 (2020) 54-56. C. E. Lapple and C. B. Shepherd, Ind. Eng. Chem. 32(5) (1940) 605-617. A. Terfous, A. Hazzab, A. Ghenaim, Powder Technol. 239 (2013) 12-20.

---

## Author Comment (AC1) · 14 Nov 2020

Referee #1 (14 November 2020) rebuttal

âŰž Kramer, O.J.I., de Moel, P.J., Raaghav, S.K.R., Baars, E.T., van Vugt, W.H., Breugem, W.-P., Padding, J.T., and van der Hoek, J.P, Can terminal settling velocity and drag of natural particles in water ever be predicted accurately?, Drinking Water Engineering Science Journal, Discussion, https://doi.org/10.5194/dwes-2020-30, under review, 2020

This paper is a very detailed discussion of the spread of 3,629 terminal settling ex-

periments and of related literature data. Scientific sustained arguments leads to the conclusion that new advanced research is needed to improve the prediction accuracy for settling velocity, drag coefficient and terminal Reynolds number of non-spherical particles such as drinking water related particles. The quality of the paper has to be upgraded by the following minor revisions:

Dear reviewer, On behalf of all authors, may I thank you very for your willingness to assess this article. The comments were highly useful and have helped to improve the article. In this rebuttal, all comments are addressed (in green). Kind regards, Onno Kramer

- Line 87: according to: : :. Replace Newton by Clift. Adjusted.

- Line 116: add: Nian-Sheng Cheng 1997 Terfous et al. 2013. Goossens 2020. Added to reference list. Note: Goossens: year is 2019 according to Powder Technology.

- Line 190-191: ""standard drag curve", ADD (Lapple-Shepherd 1940) Added tot reference list.

- Line 290: this definition of the Galileo number has to be repeated in the list of symbols. comment: this definition is peculiar as in a standard way the square root is omitted. Nomenclature updated with the dimensionless numbers e.g.: Ga=√Ar=√(gãĂŰd_pãĂŮˆ3 _f |_p-_f |/$\eta$ˆ2 ) etc. The Galileo number without the root is the Archimedes number Ar. In general, both can be used. The main reason to use Ga is that the regime map by (Jenny et al., 2004) and the one of (Zhou and Dušek, 2004) used as an important reference in the present work (Fig. 10 in the main article) uses Ga and not Ar. Furthermore, most of the literature reporting the regime maps pertaining to the instabilities of falling/rising spheres use Ga and not Ar (see for e.g. Jenny et al., 2004, Zhou and Dušek, 2004, Veldhuis and Biesheuvel, 2007). Hence, in the regime map they report (x-axis) as Ga. So, we were consistent with them.

Added to the article: Note: The majority of literature which addresses path instabilities

use the Galileo number based on the regime map (Jenny et al., 2004), and not the Archimedes number (Karamanev, 1996).

- Line 325: replace "estimated" by "experimental". Adjusted.

- List of symbols: are not used in the paper and have to be removed of the list: A,b,c, Ar As ct di dp dsi Eh50 E1.50 Ew50 Fb Fd Fg Fp k Symm UC x Greek: likewise: : :.. Redundant symbols removed. Note: these symbols were used in the Supplementary Material.

- List of symbols: Are to be defined: Cd =definition of eq. (4) Cd' _ Cd dp is volume-equivalent particle diameter Ga = definition of eq. (5) Ret = definition of eq. (2) Adjusted: equations added.

- References: ADD: Nian-Sheng Cheng, J. Hydraulic Eng. February 1997,149-152 Walter R.A. Goossens, Powder Technology 362 (2020) 54-56. C. E. Lapple and C. B. Shepherd, Ind. Eng. Chem. 32(5) (1940) 605-617. A. Terfous, A. Hazzab, A. Ghenaim, Powder Technol. 239 (2013) 12-20. Added tot reference list.

In addition: the following textual changes has been made in:

DWES Kramer 2020 Article - Manuscript.docx - fig 12 green was lost after making a PDF - pellet softening -> pellet-softening - Fractionated ipv fractionised - reuse -> re-use - Camp, 1852 -> 1946

DWES Kramer 2020 Article - Supplementary materials.docx - experimental set-up: wrong (Haynes) ref. removed

Please also note the supplement to this comment:
https://dwes.copernicus.org/preprints/dwes-2020-30/dwes-2020-30-AC1-supplement.pdf

---

## Referee Comment (RC2) · Anonymous Referee #2 · 15 Nov 2020

This paper pulls together old data and models and combines them with new data and new analyses related to predicting the terminal settling velocity of particles in a water treatment setting. I think the paper is valuable, accurate, and should be published.

I thought the paper could have been a little more streamlined and straight-forward as there was an element of trying to publish all of their materials on this topic in one place (which I think would be a good thing). However, I was not able to identify any significant strategies for removing materials or reorganizing content. I will provide a list of minor suggested revisions below.

Line 41: Date of "Camp" reference should be 1946 (not 1852).

Line 68: Sphericity is defined later in the paper, but it could be defined here for clarity.

Line 73: Tense changes unnecessarily from "were investigated" to "will investigate"

Line 75: would be good to include a few specific examples of the models being discussed unless you are referring to all of them.

Line 114: This sounds like the solutions must be numerically approximated, which sounds negative because the solutions would be time-consuming. Please verify/clarify.

Line 196: It would be good to point out the ratio of the particle size (d) to the column diameter (D), which I think is 10mm/57mm (but there are 2 columns listed in the methods).

Figure 5: I'm less than clear on exactly what the error bars represent. Are the plotted values the average/accepted values with the error bars representing calculated changes in each parameter based on the uncertainty of the model's input parameters listed in Table 4. It is not clear why the error bars vary so much in magnitude given that all of the equations use the same variables.

Line 238: "according" should be changed to "accordingly" or "according to"... unclear as is.

Line 241: does "settle horizontally" refer to the direction of settling or the orientation of the particle?

Figures 910: It is not clear why half of each graph shows density ratios less than 1. It seems these particles would float (instead of settle) and no particles appear to fit into this range. In Fig 10, the "zig zag" region only occurs where there are no particles, and the boundaries of the "chaotic" region are unclear. Is it everywhere that is white or only below a density ratio of 1. Needs revision.

Lines 406-407: Please verify that a drag coefficient increase from 1.2 to 1.7 ( 42

Line 452: This might be a good place to recommend a model or two that do predict

more accurately with more complexity?

Lines 453-459: I am not certain the authors make a strong enough case for the need of greater accuracy in predicting terminal settling velocities in water treatment applications. A little extra explanation or a concrete example here would be helpful. What would result if my terminal velocity calculations were off by 20 percent? Is a 20-30

Line 463: a more specific approach or method is requested here for the use of "more morphological properties" to include which properties and/or which models. It would be even better to be quantitative here. How much more accurate would the model be with these properties included.

The last section of the conclusion is a little vague. While the authors seem to know which model(s) are best for spherical particles and non-spherical particles, I do not think this paper is ready to publish until that information is shared with the reader in the conclusion. It would be even better to share the expected level of accuracy of the predictions for each... at least for this softening process in the narrow range of conditions in terms of Ga numbers (or similar).

---

## Author Comment (AC2) · 16 Nov 2020

Referee #2 (15 November 2020) rebuttal

? Kramer, O.J.I., de Moel, P.J., Raaghav, S.K.R., Baars, E.T., van Vugt, W.H., Breugem, W.-P., Padding, J.T., and van der Hoek, J.P., Can terminal settling velocity and drag of natural particles in water ever be predicted accurately?, Drinking Water Engineering Science Journal, Discussion, https://doi.org/10.5194/dwes-2020-30, under review, 2020

This paper pulls together old data and models and combines them with new data and

new analyses related to predicting the terminal settling velocity of particles in a water treatment setting. I think the paper is valuable, accurate, and should be published. I thought the paper could have been a little more streamlined and straight-forward as there was an element of trying to publish all of their materials on this topic in one place (which I think would be a good thing). However, I was not able to identify any significant strategies for removing materials or reorganizing content. I will provide a list of minor suggested revisions below.

Dear reviewer, On behalf of all authors, may I thank you very for your willingness to assess this article. The comments were highly useful and have helped to improve the article. In this rebuttal, all comments are addressed (in green). Kind regards, Onno Kramer

- Line 41: Date of "Camp" reference should be 1946 (not 1852). Adjusted. Indeed this was caused by a mistake in a reference manager field (Mendeley).

- Line 68: Sphericity is defined later in the paper, but it could be defined here for clarity. Adjusted. Defining sphericity after line 68 makes sense. The sphericity of a particle is the ratio of the surface area of a sphere with the same volume as the given particle to the surface area of the particle.

- Line 73: Tense changes unnecessarily from "were investigated" to "will investigate" Adjusted. Aspects such as natural variations in fluid and particle properties, the degree of polydispersity and other factors that influence the terminal settling velocity will be investigated in this work.

- Line 75: would be good to include a few specific examples of the models being discussed unless you are referring to all of them. Adjusted. I added some articles where the authors were mainly focused on the prediction accuracy of their proposed model but did not thoroughly discuss the data spread around predicted average values. An alternative solution is to refer to table 1, but I respect the opinion of the reviewer. In this work, we will investigate the amount and the causes of this spread, something which

is hugely underexposed in the popular and often cited prediction models presented in the literature e.g. (Cheng, 1997); (Khan and Richardson, 1987); (Brown and Lawler, 2003); (Zhiyao et al., 2008); (Barati and Neyshabouri, 2018).

- Line 114: This sounds like the solutions must be numerically approximated, which sounds negative because the solutions would be time-consuming. Please verify/clarify. Explanation: to be able to calculate the terminal settling velocity, a drag relation is needed. For instance the drag valid for the Stokes regime belonging to creeping or viscous flow.

CD=24/Ret

Now we need:

Ret=(rho_f dp vt)/mu

And:

CD=(4/3) (g dp |rho_p-rho_f|)/(vt^2 rho_f)

This will directly lead to the well-known Stokes equation:

vt=(1/18) (g dp^2 (rho_p - rho_f))/mu

Analytical expressions are also possible to derive for the Newton regime, belonging to the inertial regime CD=0.44 and occasionally for the transitional regime CD=10/?(Re_t ). But the literature provides a dazzling collection of much more complex expressions which are unsuitable to directly calculate the terminal settling velocity. One of the most familiar equation states is that by Brown (2003):

CD=24/Ret (1+0.15 Ret^0.681)+0.407/(1+8710/Ret)

Now, a numerical method is needed, such as the Bolzano's numerical intermediate value theorem. Please see Supplementary Material: Section: 5.4 Brown–Lawler model. This explanation is not included in the article. I hope the reviewer will acknowledge this.

- Line 196: It would be good to point out the ratio of the particle size (d) to the column diameter (D), which I think is 10mm/57mm (but there are 2 columns listed in the methods). Adjusted. In addition, in literature, wall effects are often not precisely defined by the authors. Commonly, they indicate the retardation of the velocity, as is discussed in the manuscript, but do not mention the particle moving towards the wall which also affects the settling velocity. Exceptional outliers are wetted-GAC Norit ROW 0.8 Supra grains (rods), due to particle rotation and their delayed settling behaviour, and the 10 mm glass beads, due to wall effects: (dp:D=10:57). Please see Supplementary Material: Section: 4 Wall-effects

- Figure 5: I'm less than clear on exactly what the error bars represent. Are the plotted values the average/accepted values with the error bars representing calculated changes in each parameter based on the uncertainty of the model's input parameters listed in Table 4. It is not clear why the error bars vary so much in magnitude given that all of the equations use the same variables. Dear reviewer, The symbols (((( do indeed represent the average values, based on large data sets. The error bars represent the uncertainty of these values. The larger the error bars, the greater the uncertainty. Please see Supplementary Material, Section 6 (and 7) Uncertainty analysis, where we quantified these error bars based on particle polydispersity, the degree of irregularities, wall effects, natural variations in particle density, temperature variations (fluid density and viscosity), linear thermal expansion, path instabilities (orientations) and minor decisive aspects such as surface roughness, gravity (to be consistent) and, last but not least, measurement error caused by the laboratory examiner. The table below shows the basic equation for the error bars. All derived equation can be found in the Supplementary Material 6.2 Overview uncertainty analysis equations and contribution to error. Finally, these results coincide with our personal observation during our countless experiments.

Table 18 and Table 19 please see attachment (Uncertainty analysis equations)

- Line 238: "according" should be changed to "accordingly" or "according to": : : unclear as is. Adjusted.

- Line 241: does "settle horizontally" refer to the direction of settling or the orientation of the particle? This refers to the orientation of the particles.

- Figures 9+10: It is not clear why half of each graph shows density ratios less than 1. It seems these particles would float (instead of settle) and no particles appear to fit into this range. In Fig 10, the "zig zag" region only occurs where there are no particles, and the boundaries of the "chaotic" region are unclear. Is it everywhere that is white or only below a density ratio of 1. Needs revision. Adjusted. This state diagram is based on literature which addresses path instabilities (Jenny et al., 2004); (Veldhuis and Biesheuvel, 2007); (Zhou and Dušek, 2015). They studied, besides, settling particles also floating particle, where the density ratios are less than 1. In addition, this also carried out by Raaghav, 2019) who is co-author of this research. This regime map is a bit complex. Therefore figures 9 and 10 are updated. A red line indicated the boundary between steady oblique and chaotic regime. In addition floating and settling regime is indicated.

Figure 9 State diagram. Galileo number Ga versus specific gravity number ?Âă? with examined particles Figure 10 Path trajectory regime plot according (Zhou and Dušek, 2015; Raaghav, 2019) (zoomed area). The chaotic regime applies to the right of the red line Please see attachment

- Lines 406-407: Please verify that a drag coefficient increase from 1.2 to 1.7 ( 42 Explanation: let's take this drag standard equation below for sand grains dp=0.8 [mm] with a particle density rho_p= 2,675 [kg/m3] at 13 [°C]. A sphericity ?=1 leads to CD = 1.18 and a sphericity ?=0.7 to CD = 1.73.

$CD=(4/3) (gdp |rho\_p-rho\_f|)/(vt^2 rho\_f)$

- Line 452: This might be a good place to recommend a model or two that do predict

more accurately with more complexity? This is exactly the reason for writing this article. Based on table 3, one could say that these models are the most accurate (lowest NRMSE): Haider–Levenspiel (1989) 8.8% vt and 20.0% in CD Brown–Lawler (2003) 9.0% vt and 17.1% in CD Using other statistical measures will lead to other 'winners' and also depending on which parameter vt or CD is examined. A 'general' accurate model which uses particle shape descriptors covering the whole Reynolds regime in the Standard Drag Curve does not exist. For this reason it is more a personal preference depending on the distinct regime which model is most suitable. But to answer the reviewer, we would suggest the Brown–Lawler equation since it is one of the most cited equation.

- Lines 453-459: I am not certain the authors make a strong enough case for the need of greater accuracy in predicting terminal settling velocities in water treatment applications. A little extra explanation or a concrete example here would be helpful. What would result if my terminal velocity calculations were off by 20 percent? Is a 20-30 The following example explains what could happen when the estimated terminal settling velocity is 20-30% off. Let us take calcite pellets dp = 1.2 [mm] which are typical in pellet-softening (fluidised bed) reactors (Graveland et al., 1983) with a particle density rho_p = 2,711 [kg/m3] at 12 [°C]. Richardson-Zaki as a voidage prediction model in full-scale pellets-softening reactors was proposed by (van Schagen, 2009) based on the terminal settling velocity. Based on Brown–Lawler (2003), the terminal settling velocity is vt = 0.18 [m/s]. Accordingly, the particle terminal Reynolds number can be calculated: Re_t = 174. Based on (Richardson and Zaki, 1954) the empirical index becomes n_RZ = 2.63 from which the voidage can be determined: ? = 0.44. This is an expected value since pellet-softening works in the vicinity of the incipient state for the maximum crystallisation surface area. If 20-30% error in vt happens, the highest predicted voidage is ? = 0.49 and the lowest voidage ? = 0.40. The highest voidage means that the softening performance decreases slightly due to less available surface area, but the lowest voidage means that a fixed bed state (? < 0.40) might occur, which leads to unwanted clogging of CaCO3 which is very harmful for the process and caused

process shut-down. We have added a warning of this possible consequence to section 3.7. Other examples could be given to elucidate the deteriorated flushing process of wash towers and backwash procedures due to poor settling velocity estimations. The message of this article is to show that there is a significant spread of data, which causes can be identified and which are relevant. Although it is interesting to explain in detail what the consequences might be, this in fact is another scope and for next research topics.

- Line 463: a more specific approach or method is requested here for the use of "more morphological properties" to include which properties and/or which models. It would be even better to be quantitative here. How much more accurate would the model be with these properties included. Numerous works exist about particle identification (Seville and Yu, 2016), and although a large number of shape factors, morphological properties and descriptors are proposed (Allen, 1990), there is no universal agreement on how to define particle shape and there is no agreement on how to correctly include the influence of irregularly shaped particles. The size and shape of the particles has important implications for, for instance, filter design (Crittenden et al., 2012), but there is no easy way to account for this. Explaining the possible effects of different shape parameters in detail is a complete new substantial research question and, although very interesting, would at this point be guesswork and therefore does not provide new insights for this paper.

- The last section of the conclusion is a little vague. While the authors seem to know which model(s) are best for spherical particles and non-spherical particles, I do not think this paper is ready to publish until that information is shared with the reader in the conclusion. It would be even better to share the expected level of accuracy of the predictions for each: : : at least for this softening The last section is used to trigger other researchers, in particular in the field of CFD, to use advanced models to increase the prediction accuracy. We do not know which model(s) are the most suitable for general purposes. Regarding pellet-softening, 5% accuracy (instead of 20-30%) using for instance the Brown-Lawler model, will decrease the voidage prediction error to 2%. In Section 3.7 Consequences of uncertainty in settling velocity for water treatment processes, we mentioned: There is no model for the prediction accuracy for terminal settling velocity and drag coefficient, that covers the wide range of differences in particle properties with a low prediction inaccuracy (<1%). Table 3 shows considerably larger errors, so a lot of work needs to be done to improve models to approach 1%. Therefore, a large uncertainty in vt has a considerable effect on the voidage prediction, for instance leading to a fixed bed state where a fluidised bed was expected. In this work, we have explicitly shown the causes of uncertainty in vt.

Please also note the supplement to this comment:
https://dwes.copernicus.org/preprints/dwes-2020-30/dwes-2020-30-AC2-supplement.pdf

———————————————

[Figure]

*Figure 9*   State diagram. Galileo number
Ga versus specific gravity number
$\bar{\rho}$ with examined particles

*Figure 10*   Path trajectory regime plot
according (Zhou and Dušek, 2015;
Raaghav, 2019) (zoomed area).
The chaotic regime applies to the
right of the red line

**Fig. 1.**

**Supplement:**

► Kramer, O.J.I., de Moel, P.J., Raaghav, S.K.R., Baars, E.T., van Vugt, W.H., Breugem, W.-P., Padding, J.T., and van der Hoek, J.P., Can terminal settling velocity and drag of natural particles in water ever be predicted accurately?, Drinking Water Engineering Science Journal, Discussion, https://doi.org/10.5194/dwes-2020-30, under review, 2020

This paper pulls together old data and models and combines them with new data and new analyses related to predicting the terminal settling velocity of particles in a water treatment setting. I think the paper is valuable, accurate, and should be published. I thought the paper could have been a little more streamlined and straight-forward as there was an element of trying to publish all of their materials on this topic in one place (which I think would be a good thing). However, I was not able to identify any significant strategies for removing materials or reorganizing content. I will provide a list of minor suggested revisions below.

Dear reviewer,
On behalf of all authors, may I thank you very for your willingness to assess this article. The comments were highly useful and have helped to improve the article. In this rebuttal, all comments are addressed (in green).
Kind regards,
Onno Kramer

- Line 41: Date of "Camp" reference should be 1946 (not 1852).
Adjusted. Indeed this was caused by a mistake in a reference manager field (Mendeley).

- Line 68: Sphericity is defined later in the paper, but it could be defined here for clarity.
Adjusted. Defining sphericity after line 68 makes sense.
The sphericity of a particle is the ratio of the surface area of a sphere with the same volume as the given particle to the surface area of the particle.

- Line 73: Tense changes unnecessarily from "were investigated" to "will investigate"
Adjusted.
Aspects such as natural variations in fluid and particle properties, the degree of polydispersity and other factors that influence the terminal settling velocity will be investigated in this work.

- Line 75: would be good to include a few specific examples of the models being discussed unless you are referring to all of them.
Adjusted. I added some articles where the authors were mainly focused on the prediction accuracy of their proposed model but did not thoroughly discuss the data spread around predicted average values. An alternative solution is to refer to table 1, but I respect the opinion of the reviewer.

In this work, we will investigate the amount and the causes of this spread, something which is hugely underexposed in the popular and often cited prediction models presented in the literature e.g. (Cheng, 1997); (Khan and Richardson, 1987); (Brown and Lawler, 2003); (Zhiyao et al., 2008); (Barati and Neyshabouri, 2018).

- Line 114: This sounds like the solutions must be numerically approximated, which sounds negative because the solutions would be time-consuming. Please verify/clarify.
Explanation: to be able to calculate the terminal settling velocity, a drag relation is needed. For instance the drag valid for the Stokes regime belonging to creeping or viscous flow.

$$C_D = \frac{24}{Re_t}$$

Now we need:

$$Re_t = \frac{\rho_f d_p v_t}{\eta}$$

And:

$$C_D = \frac{4}{3} \frac{g d_p |\rho_p - \rho_f|}{v_t^2 \rho_f}$$

This will directly lead to the well-known Stokes equation:

$$v_t = \frac{1}{18} \frac{g d_p^2 (\rho_p - \rho_f)}{\eta}$$

Analytical expressions are also possible to derive for the Newton regime, belonging to the inertial regime $C_D = 0.44$ and occasionally for the transitional regime $C_D = 10/\sqrt{Re_t}$. But the literature provides a dazzling collection of much more complex expressions which are unsuitable to directly calculate the terminal settling velocity. One of the most familiar equation states is that by Brown (2003):

$$C_D = \frac{24}{Re_t}\left(1 + 0.15 Re_t^{0.681}\right) + \frac{0.407}{1 + \frac{8710}{Re_t}}$$

Now, a numerical method is needed, such as the Bolzano's numerical intermediate value theorem.
Please see Supplementary Material: Section: 5.4 Brown–Lawler model.

This explanation is not included in the article. I hope the reviewer will acknowledge this.

- Line 196: It would be good to point out the ratio of the particle size (d) to the column diameter (D), which I think is 10mm/57mm (but there are 2 columns listed in the methods).
Adjusted. In addition, in literature, wall effects are often not precisely defined by the authors. Commonly, they indicate the retardation of the velocity, as is discussed in the manuscript, but do not mention the particle moving towards the wall which also affects the settling velocity.
Exceptional outliers are wetted-GAC Norit ROW 0.8 Supra grains (rods), due to particle rotation and their delayed settling behaviour, and the 10 mm glass beads, due to wall effects: $(d_p : D = 10 : 57)$.
Please see Supplementary Material: Section: 4 Wall-effects

- Figure 5: I'm less than clear on exactly what the error bars represent. Are the plotted values the average/accepted values with the error bars representing calculated changes in each parameter based on the uncertainty of the model's input parameters listed in Table 4. It is not clear why the error bars vary so much in magnitude given that all of the equations use the same variables.
Dear reviewer, The symbols △◇□○ do indeed represent the average values, based on large data sets. The error bars represent the uncertainty of these values. The larger the error bars, the greater the uncertainty. Please see Supplementary Material, Section 6 (and 7) Uncertainty analysis, where we quantified these error bars based on particle polydispersity, the degree of irregularities, wall effects, natural variations in particle density, temperature variations (fluid density and viscosity), linear thermal expansion, path instabilities (orientations) and minor decisive aspects such as surface roughness, gravity (to be consistent) and, last but not least, measurement error caused by the laboratory examiner.
The table below shows the basic equation for the error bars. All derived equation can be found in the Supplementary Material 6.2 Overview uncertainty analysis equations and contribution to error.
Finally, these results coincide with our personal observation during our countless experiments.

*Table 18* *Uncertainty analysis equations*

| Variable | Equation | Eq. nr. |
|---|---|---|
| Terminal Reynolds number | $\delta Re_t = \sqrt{\left(\dfrac{\partial Re_t}{\partial d_p}\delta d_p\right)^2 + \left(\dfrac{\partial Re_t}{\partial v_T}\delta v_T\right)^2 + \left(\dfrac{\partial Re_t}{\partial v_t}\delta v_t\right)^2}$ | (69) |
| Drag coefficient | $\delta C_D = \sqrt{\left(\dfrac{\partial C_D}{\partial g}\delta g\right)^2 + \left(\dfrac{\partial C_D}{\partial d_p}\delta d_p\right)^2 + \left(\dfrac{\partial C_D}{\partial \rho_p}\delta \rho_p\right)^2 + \left(\dfrac{\partial C_D}{\partial \rho_f}\delta \rho_f\right)^2 + \left(\dfrac{\partial C_D}{\partial v_t}\delta v_t\right)^2}$ | (70) |

Particle density

$$\delta \rho_p = \sqrt{\left(\frac{\partial \rho_p}{\partial m_p}\delta m_p\right)^2 + \left(\frac{\partial \rho_p}{\partial d_p}\delta d_p\right)^2}$$

(71)

Terminal settling velocity [1)]

$$\delta v_t = \sqrt{\left(\frac{\partial v_t}{\partial L}\delta L\right)^2 + \left(\frac{\partial v_t}{\partial t}\delta t\right)^2} \qquad \delta t = c_0 + c_1 e^{-t}$$

(72)
* * *
[1)] Human response time inaccuracy correction is given in §6.9.4

**Table 19** *Uncertainty analysis equations*

| Variable | Term | Equation | Eq. nr. (contribution to error) |
|---|---|---|---|
| Terminal Reynolds number (Eq. 67) | 1st | $\dfrac{\partial Re_t}{\partial d_p} = \dfrac{v_t}{v_T}$ | (73) |
| | 2nd | $\dfrac{\partial Re_t}{\partial T} = \dfrac{d_p v_t}{c_6}\left(-\dfrac{c_7 ln10(1+\alpha\Delta T)}{(T+c_8)^2 10^{c_7/(T+c_8)}} - \dfrac{\alpha}{10^{c_7/(T+c_8)}}\right)$ | |
| | 3rd | $\dfrac{\partial Re_t}{\partial v_t} = \dfrac{1}{v_T}\left(d_p - \dfrac{c_2 d_p^2}{D}\right)$ | (74) |
| Drag coefficient (Eq. 68) | 1st | $\dfrac{\partial C_D}{\partial g} = \dfrac{4}{3}\dfrac{d_p}{v_t^2}\left(\dfrac{\rho_p}{\rho_f}-1\right)$ | |
| | 2nd | $\dfrac{\partial C_D}{\partial d_p} = \dfrac{\frac{4}{3}\frac{g}{v_t^2}\left(\frac{\rho_p}{\rho_f}-1\right)}{\left(d_p^{-\frac{1}{2}}-c_2\dfrac{d_p^{\frac{1}{2}}}{D}\right)^3}\left(d_p^{-\frac{3}{2}}+\dfrac{c_2}{D}d_p^{-\frac{1}{2}}\right)$ | (75) |
| | 3rd | $\dfrac{\partial C_D}{\partial \rho_p} = \dfrac{4}{3}\dfrac{g d_p}{v_t^2 \rho_f}$ | |
| | 4th | $\dfrac{\partial C_D}{\partial T} = \dfrac{4}{3}\dfrac{g d_p}{v_t^2}\left(\rho_p\left(\dfrac{c_3 e^{c_3 T}}{c_4 - c_5 T^2}+\dfrac{2c_5 T e^{c_3 T}}{(c_4 - c_5 T^2)^2}\right) - c_3 e^{c_3 T}\right)$ | (76) |
| | 5th | $\dfrac{\partial C_D}{\partial v_t} = -\dfrac{8}{3}\dfrac{g d_p}{v_t^3}\dfrac{\left(\frac{\rho_p}{\rho_f}-1\right)}{\left(1-c_2\dfrac{d_p}{D}\right)^2}$ | |
| Particle density (Eq. 71) | 1st | $\dfrac{\partial \rho_p}{\partial m_p} = \dfrac{6}{\pi d_p^3}$ | (77) |
| | 2nd | $\dfrac{\partial \rho_p}{\partial d_p} = -\dfrac{18 m_p}{\pi d_p^4}$ | |
| Terminal settling velocity (Eq. 72) | 1st | $\dfrac{\partial v_t}{\partial L} = \dfrac{1}{t}$ | (78) |
| | 2nd | $\dfrac{\partial v_t}{\partial t} = -\dfrac{L}{t^2}$ | |

- Line 238: "according" should be changed to "accordingly" or "according to": : : unclear as is.
Adjusted.

- Line 241: does "settle horizontally" refer to the direction of settling or the orientation of the particle?
This refers to the orientation of the particles.

- Figures 9+10: It is not clear why half of each graph shows density ratios less than 1. It seems these particles would float (instead of settle) and no particles appear to fit into this range. In Fig 10, the "zig zag" region only occurs where there are no particles, and the boundaries of the "chaotic" region are unclear. Is it everywhere that is white or only below a density ratio of 1. Needs revision.
Adjusted. This state diagram is based on literature which addresses path instabilities (Jenny et al., 2004); (Veldhuis and Biesheuvel, 2007); (Zhou and Dušek, 2015). They studied, besides, settling particles also floating particle, where the density ratios are less than 1. In addition, this also carried out by Raaghav, 2019) who is co-author of this research.
This regime map is a bit complex. Therefore figures 9 and 10 are updated. A red line indicated the boundary between steady oblique and chaotic regime. In addition floating and settling regime is indicated.

[Figure]

**Figure 9**    State diagram. Galileo number $Ga$ versus specific gravity number $\bar{\rho}$ with examined particles

[Figure]

**Figure 10**    Path trajectory regime plot according (Zhou and Dušek, 2015; Raaghav, 2019) (zoomed area). The chaotic regime applies to the right of the red line

- Lines 406-407: Please verify that a drag coefficient increase from 1.2 to 1.7 ( 42

Explanation: let's take this drag standard equation below for sand grains $d_p$=0.8 [mm] with a particle density $\rho_p$= 2,675 [kg/m³] at 13 [ºC].
A sphericity Φ=1 leads to $C_D$ = 1.18 and a sphericity Φ=0.7 to $C_D$ = 1.73.

$$C_D = \frac{4}{3}\frac{g d_p |\rho_p - \rho_f|}{v_t{}^2 \rho_f}$$

- Line 452: This might be a good place to recommend a model or two that do predict more accurately with more complexity?
This is exactly the reason for writing this article. Based on table 3, one could say that these models are the most accurate (lowest NRMSE):
Haider–Levenspiel (1989) 8.8% $v_t$ and 20.0% in $C_D$
Brown–Lawler (2003) 9.0% $v_t$ and 17.1% in $C_D$
Using other statistical measures will lead to other 'winners' and also depending on which parameter $v_t$ or $C_D$ is examined.
A 'general' accurate model which uses particle shape descriptors covering the whole Reynolds regime in the Standard Drag Curve does not exist.
For this reason it is more a personal preference depending on the distinct regime which model is most suitable.
But to answer the reviewer, we would suggest the Brown–Lawler equation since it is one of the most cited equation.

- Lines 453-459: I am not certain the authors make a strong enough case for the need of greater accuracy in predicting terminal settling velocities in water treatment applications. A little extra explanation or a concrete example here would be helpful. What would result if my terminal velocity calculations were off by 20 percent? Is a 20-30
The following example explains what could happen when the estimated terminal settling velocity is 20-30% off. Let us take calcite pellets $d_p$ = 1.2 [mm] which are typical in pellet-softening (fluidised bed) reactors (Graveland et al., 1983) with a particle density $\rho_p$ = 2,711 [kg/m³] at 12 [ºC].
Richardson-Zaki as a voidage prediction model in full-scale pellets-softening reactors was proposed by (van Schagen, 2009) based on the terminal settling velocity.
Based on Brown–Lawler (2003), the terminal settling velocity is $v_t$ = 0.18 [m/s]. Accordingly, the particle terminal Reynolds number can be calculated: $Re_t$ = 174. Based on (Richardson and Zaki, 1954) the empirical index becomes $n_{RZ}$ = 2.63 from which the voidage can be determined: $\varepsilon$ = 0.44. This is an expected value since pellet-softening works in the vicinity of the incipient state for the maximum crystallisation surface area. If 20-30% error in $v_t$ happens, the highest predicted voidage is $\varepsilon$ = 0.49 and the lowest voidage $\varepsilon$ = 0.40. The highest voidage means that the softening performance decreases slightly due to less available surface area, but the lowest voidage means that a fixed bed state ($\varepsilon$ < 0.40) might occur, which leads to unwanted clogging of $CaCO_3$ which is very harmful for the process and caused process

shut-down. We have added a warning of this possible consequence to section 3.7.
Other examples could be given to elucidate the deteriorated flushing process of wash towers and backwash procedures due to poor settling velocity estimations. The message of this article is to show that there is a significant spread of data, which causes can be identified and which are relevant. Although it is interesting to explain in detail what the consequences might be, this in fact is another scope and for next research topics.

- Line 463: a more specific approach or method is requested here for the use of "more morphological properties" to include which properties and/or which models. It would be even better to be quantitative here. How much more accurate would the model be with these properties included.
Numerous works exist about particle identification (Seville and Yu, 2016), and although a large number of shape factors, morphological properties and descriptors are proposed (Allen, 1990), there is no universal agreement on how to define particle shape and there is no agreement on how to correctly include the influence of irregularly shaped particles. The size and shape of the particles has important implications for, for instance, filter design (Crittenden et al., 2012), but there is no easy way to account for this. Explaining the possible effects of different shape parameters in detail is a complete new substantial research question and, although very interesting, would at this point be guesswork and therefore does not provide new insights for this paper.

- The last section of the conclusion is a little vague. While the authors seem to know which model(s) are best for spherical particles and non-spherical particles, I do not think this paper is ready to publish until that information is shared with the reader in the conclusion. It would be even better to share the expected level of accuracy of the predictions for each: : : at least for this softening
The last section is used to trigger other researchers, in particular in the field of CFD, to use advanced models to increase the prediction accuracy.
We do not know which model(s) are the most suitable for general purposes. Regarding pellet-softening, 5% accuracy (instead of 20-30%) using for instance the Brown-Lawler model, will decrease the voidage prediction error to 2%.
In Section 3.7 Consequences of uncertainty in settling velocity for water treatment processes, we mentioned: *There is no model for the prediction accuracy for terminal settling velocity and drag coefficient, that covers the wide range of differences in particle properties with a low prediction inaccuracy (<1%)*. Table 3 shows considerably larger errors, so a lot of work needs to be done to improve models to approach 1%.
Therefore, a large uncertainty in $v_t$ has a considerable effect on the voidage prediction, for instance leading to a fixed bed state where a fluidised bed was expected. In this work, we have explicitly shown the causes of uncertainty in $v_t$.
https://doi.org/10.5194/dwes-2020-